# MrSteve: Instruction-Following Agents in Minecraft with What-Where-When Memory

**Junyeong Park**[1*], **Junmo Cho**[1*], **Sungjin Ahn**[1,2]
[1]KAIST & [2]New York University

## Abstract

Significant advances have been made in developing general-purpose embodied AI in environments like Minecraft through the adoption of LLM-augmented hierarchical approaches. While these approaches, which combine high-level planners with low-level controllers, show promise, low-level controllers frequently become performance bottlenecks due to repeated failures. In this paper, we argue that the primary cause of failure in many low-level controllers is the absence of an episodic memory system. To address this, we introduce MrSteve (**M**emory **R**ecall Steve), a novel low-level controller equipped with Place Event Memory (PEM), a form of episodic memory that captures what, where, and when information from episodes. This directly addresses the main limitation of the popular low-level controller, Steve-1. Unlike previous models that rely on short-term memory, PEM organizes spatial and event-based data, enabling efficient recall and navigation in long-horizon tasks. Additionally, we propose an Exploration Strategy and a Memory-Augmented Task Solving Framework, allowing agents to alternate between exploration and task-solving based on recalled events. Our approach significantly improves task-solving and exploration efficiency compared to existing methods. We will release our code and demos on the project page: https://sites.google.com/view/mr-steve.

## 1 Introduction

The emergence of large-scale foundation models has driven significant advances in developing general-purpose embodied AI agents capable of generalizing across a broad spectrum of tasks in complex, open, and real-world-like environments (Johnson et al., 2016; Guss et al., 2019; Fan et al., 2022b; Hafner, 2022; Albrecht et al., 2022; Voudouris et al., 2023). While simulating such environments for effective learning and evaluation remains a major challenge, Minecraft has become a leading testbed, offering a demanding, open-ended environment with rich interaction possibilities. Its procedurally generated world presents agents with challenges like exploration, resource management, tool crafting, and survival, all requiring advanced decision-making and long-horizon planning. For instance, the task of obtaining a diamond 💎 requires agents to locate diamond ore 🟦, and craft an iron pickaxe ⛏. This process involves finding, mining, and refining iron ore 🟫, requiring the agent to execute detailed long-term planning over roughly 24,000 environmental steps (Li et al., 2024).

Solving such tasks through Reinforcement Learning (RL) approaches from scratch is nearly infeasible; however, recent LLM-augmented hierarchical methods have demonstrated a promising avenue (Huang et al., 2022a;b; Wang et al., 2023a). These methods feature a division between high-level planners and low-level controller policies. High-level planners, driven by Large Language Models (LLMs) or Multimodal Large Language Models (MLLMs), propose subgoals by utilizing the reasoning abilities and prior knowledge inherent in LLMs (Brown et al., 2020; Touvron et al., 2023; OpenAI, 2024). These subgoals, conveyed in textual instruction form, are then sequentially passed to a learned, instruction-following low-level controller for execution (Wang et al., 2023c;b; Li et al., 2024).

For this framework to be effective, it is essential that both the high-level planner and the low-level controller improve in tandem. However, previous research has primarily focused on enhancing high-level planning, *e.g.*, via maintaining skill library (Zhu et al., 2023; Wang et al., 2023b; Qin et al.,

---

*Equal contribution. Correspondence to Junyeong Park and Sungjin Ahn.
Contact:{jyp10987,sungjin.ahn}@kaist.ac.kr

2024; Li et al., 2024), often assuming that low-level controllers will efficiently execute the subgoals provided by the high-level planner. However, this assumption frequently does not hold in practice, and the low-level controller becomes a significant performance bottleneck (Cai et al., 2023b).

In this regard, we specifically focus on limitations in Steve-1 (Lifshitz et al., 2024), the most widely used low-level instruction-following controller framework. Steve-1 is an instruction-following policy obtained by fine-tuning the Video Pre-Training (VPT) (Baker et al., 2022) model. A primary limitation we focus on is its constrained episodic memory capability. Steve-1 is based on Transformer-XL (Dai et al., 2019), which leverages relatively short-term memory, retaining only the last 128 hidden states. Given Minecraft's simulation speed of 20Hz, this memory span amounts to only a few seconds of gameplay. While it can be increased, the quadratic complexity and FIFO-only memory structure of transformers make them significantly inefficient for long-horizon tasks.

As a result, when the agent requires information beyond this short memory span, it tends to forget past events within the episode and reverts to inefficient random exploration for each new task, consuming excessive time. For example, when given a task like "Find a Cow", the agent is unable to recall, '*I've seen it before near the river in the north*'. Ideally, a low-level agent would instead maintain an episodic memory of meaningful events and recall relevant information. See Figure 1 for more detail illustration. Moreover, Steve-1 not only lacks the ability to recall such memories but also the ability to navigate directly to the associated locations, which could help avoid unnecessary exploration. Instead, Steve-1 relies on a "go explore" instruction, randomly exploring until it stumbles upon the resource by chance. This inefficiency in executing low-level primitives is not addressed by high-level planners, which focus on optimizing the sequence of high-level skills (*i.e.*, the plan) but do not optimize the execution of the primitives themselves. We elaborate more on this in Appendix A.

In this paper, we introduce an enhanced low-level controller agent, MrSteve (**M**emory **R**ecall Steve), designed to address the limitations of Steve-1. The key innovation of MrSteve is the integration of Place Event Memory (PEM) which is the instantiation of What-Where-When Episodic Memory. While previous approaches have explored episodic memory, they primarily target high-level planners, such as building libraries of high-level skills and plans, which do not directly improve the low-level controller's performance. We argue it is essential for the low-level controller to possess memory capabilities. To address this, PEM manages memory more effectively, surpassing the limitations of the non-scalable FIFO memory found in transformers. PEM stores spatial and event-based information, allowing the agent to hierarchically organize and retrieve details about locations and events it has previously encountered. For PEM to be fully effective, the agent must also move directly to the desired location along with the ability to modulate between exploration and goal-directed navigation-and-execution, a capability lacking in Steve-1. Therefore, we introduce the second component of MrSteve: the Exploration Strategy and Memory-Augmented Task Solving Framework. Built upon the PEM structure, this framework enables the agent to alternate between exploration—when no relevant information is stored—and task-solving by recalling past events when applicable. This is made possible and effective with our new navigation policy, VPT-Nav.

Our contributions are as follows. First, we point out the limitations of Steve-1, the most widely used instruction-following controller, and show how its bottlenecks can be addressed with MrSteve. Second, we introduce Place Event Memory (PEM), a novel hierarchical memory system that organizes spatial and event-based data for efficient querying and storage, even under limited memory capacity. Third, we propose an Exploration Strategy and Task Solving Module built on PEM that enables efficient exploration while maintaining high task-solving performance in Minecraft. Last, we demonstrate that our agent significantly outperforms existing baselines in both exploration and long sequence of tasks solving. We will release the code for further research.

## 2 RELATED WORKS

**Low-Level Controllers in Minecraft** Earlier works (Guss et al., 2019; Lin et al., 2021; Mao et al., 2022; Cai et al., 2023a; Hafner et al., 2023; Zhou et al., 2024a) introduced policy models for simple tasks in Minecraft. MineCLIP (Fan et al., 2022b) leveraged text-video data to train a contrastive video-language model as a reward model, while VPT (Baker et al., 2022) was pre-trained on unlabelled videos without text-based instruction input. Steve-1 (Lifshitz et al., 2024) extended VPT by incorporating text instructions to generate low-level actions based on human demonstration data. GROOT (Cai et al., 2023b) used reference video instead of text for goal-conditioned behavior

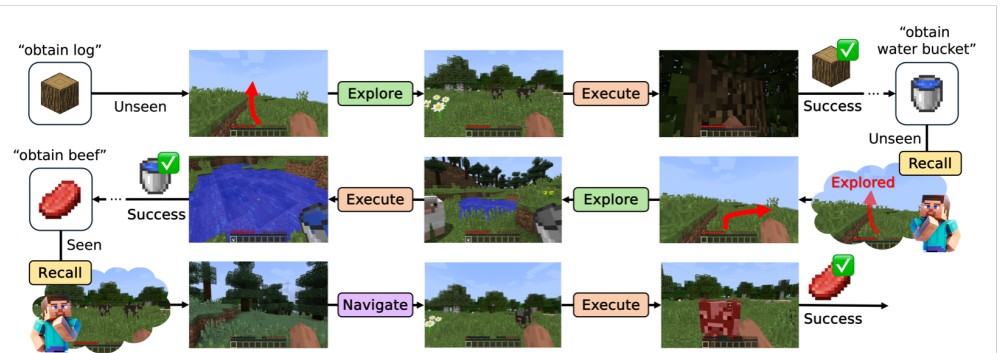

**Figure 1:** Sparse Sequential Task Solving Scenario. The first task is to *obtain a log*. The agent explores to find a tree. While searching, the agent observes a cow but continues focusing on acquiring the log. Once the log is obtained, the next task is to *obtain a water bucket*. Remembering that it already explored the forward direction while searching for the tree, the agent chooses to explore to the right. After gathering the water bucket, the final task is *obtain meat*, which can be acquired from the cow. Recalling the cow's location, the agent navigates there and completes the task by obtaining the meat. Note that each task takes a few thousand steps to achieve. This scenario highlights the significance of episodic memory for efficient exploration and task-solving in an open-ended world where task-relevant resources are sparsely distributed.

cloning. Recently, MineDreamer (Zhou et al., 2024b) leveraged Steve-1 generating subgoal images with MLLM and Diffusion based on text and current observation for improved control. However, these agents lack episodic memory, forcing agents to start new tasks from scratch. MrSteve addresses this by integrating episodic memory, making it more effective in sequential tasks.

**LLM-Augmented Agents** The development of LLMs has significantly advanced agents in Minecraft (Wang et al., 2023a;b). These works utilize pre-trained LLMs as zero-shot planners (Brown et al., 2020; Touvron et al., 2023), leveraging their powerful reasoning capabilities to generate subgoal plans or executable code. Broadly, this line of research can be divided into two approaches: one that uses LLMs for code generation to interact with the environment directly (Wang et al., 2023a; Zhu et al., 2023; Qin et al., 2024; Liu et al., 2024), and another that generates text-based subgoals which are then executed by a goal-conditioned low-level controller, such as Steve-1 or programmed heuristics (Nottingham et al., 2023; Yuan et al., 2023; Li et al., 2024). In the latter approach, to ensure LLMs focus on high-level semantic reasoning, the low-level controller must efficiently execute subgoals. While combining LLM as a high-level planner with MrSteve is one possible direction, we focus on enhancing low-level controller's capabilities based on the new type of memory in this work.

**Memory in Agents** Memory systems in agents primarily aim to retrieve robust and accurate high-level plans for long-horizon tasks (Zhang et al., 2023; Song et al., 2023; Kagaya et al., 2024; Sun et al., 2024; Shinn et al., 2024). Existing works store successful task's text instruction and its plans in language often with observations for robust retrieval, which is useful when plans for the new task already exist in memory. Voyager (Wang et al., 2023a) uses an unimodal storage of achieved skill codes in the form of text. GITM (Zhu et al., 2023) integrates text-based knowledge and memory for higher reasoning efficiency and stores entire skill codes after a goal is achieved. Recently, MP5 (Qin et al., 2024) and JARVIS-1 (Wang et al., 2023b) enhance planning by storing plans and whole multimodal observations in the abstracted memory, allowing for situation-aware retrieval, while Optimus-1 (Li et al., 2024) introduces a multimodal experience pool that summarizes all multimodal information during agent's execution of the task improving storage and retrieval efficiency. However, these memory systems store the sequence of high-level skills or plans for high-level planners, which are not optimized for low-level controllers. We address this problem with Place Event Memory.

## 3 METHOD

In this section, we describe our agent, MrSteve (**M**emory **R**ecall Steve). We begin with the problem setting, followed by step-by-step construction of our agent's main modules.

**Problem Setting** In this work, we define a sparse sequential task scenario where the agent is continuously given tasks $\{\tau_n\}_{n=1}^{\infty}$ through text instructions (*e.g.*, *Obtain water bucket)* from the

**Figure 2:** MrSteve and Place Event Memory. (a) MrSteve takes agent's position, first person view, and text instruction, and utilizes Memory Module and Solver Module to follow the instruction. (b) MrSteve leverages Place Event Memory for exploration and task execution, which stores the novel events from visited places.

environment or subgoal plans by LLM. Additionally, we assume that task-relevant resources (*e.g.*, water, cow) **rarely exist** and are **sparsely distributed** in the environment, making it essential to memorize novel events from visited places for future tasks as shown in Figure 1. When an episode begins, for every time step $t$, the agent is provided with the observation $X_t = \{i_t, l_t, t\}$, which consists of the pixel observation $i_t \in \mathbb{R}^{H \times W \times C}$, representing the first person view of the environment, the positional information $l_t = (\text{coord}_x, \text{coord}_y, \text{coord}_z, \text{yaw}, \text{pitch}) \in \mathbb{R}^5$, which denotes the agent's relative 3D position and camera angles with respect to initial position $l_0$, and time $t$.

**Instruction Following Policy** In sparse sequential task, a naive approach is to employ Steve-1 (Lifshitz et al., 2024), an instruction-following policy $\pi_{\text{Inst}}(a_t | h_t, \tau_n)$ that generates low-level controls (mouse and keyboard) in Minecraft. Here, $h_t$ is a past pixel observation sequence $i_{t-128:t}$. While past observations are processed by Transformer-XL layers in Steve-1, the model is ineffective at recalling observations from a few thousand steps ago (Lampinen et al., 2021). Additionally, the Transformer's quadratic complexity makes it significantly inefficient to process thousands of observations. This makes Steve-1 poorly suited for sparse sequential task, as it cannot recall visited places or task-relevant resources seen in the past. To address this, we propose MrSteve which stores novel events from visited places for efficient sparse sequential task-solving.

**MrSteve** is a memory-augmented instruction following policy that consists of Memory Module and Solver Module, as shown in Figure 2(a). In Memory Module, we use the memory called Place Event Memory $M_t$ that stores novel events from visited places (Figure 2(b)). Based on Place Event Memory, Mode Selector in Solver Module decides between Explore mode and Execute mode. When no task-relevant resource exists in the memory, Explore mode is selected, and the agent explores with our hierarchical exploration method. If a task-relevant resource exists in the memory, Execute mode is selected, then the agent navigates to the resource's position and executes $\pi_{\text{Inst}}$ (*i.e.*, Steve-1) to solve the task. Algorithm 1 outlines the task-solving loop of MrSteve, which repeats every fixed step or when a new task is given (More details in Appendix D). With these modules, MrSteve can efficiently explore and recall task-relevant resources from the memory to solve

---

**Algorithm 1** MrSteve Single Loop

**Require:** Memory $M_t$, and task $\tau_n$
1: $candidates \leftarrow \text{Read}(M_t, \tau_n)$
2: **if** $candidates \neq \emptyset$ **then**
3: $\quad X_t, l_t = \text{OneOf}(candidates)$
4: $\quad$ Navigate to $l_t$ with $\pi_{\text{L-Nav}}$
5: $\quad$ Execute $\tau_n$ with $\pi_{\text{Inst}}$
6: **else**
7: $\quad$ Explore with $\pi_{\text{H-Cnt}}, \pi_{\text{L-Nav}}$
8: **end if**

---

sparse sequential task in Figure 1. In the following sections, we describe how each module in MrSteve is constructed. We begin by constructing Memory Module.

## 3.1 MEMORY MODULE: CONSTRUCTION OF PLACE EVENT MEMORY

The simplest form of memory is FIFO Memory, denoted as $M_t$ with capacity $N$. At every time step, instead of storing $X_t$ in $M_t$, we can extract a semantic representation from the video $i_{t-H:t}$ with a video encoder to store an experience frame $x_t = \{e_t, l_t, t\}$, where $e_t = \text{Enc}_v(i_{t-H:t})$. For simplicity, we term $e_t$ as the video embedding at time step $t$. When the memory exceeds its capacity, the oldest frame is removed. For memory read, we calculate the cosine similarity between the task embedding $\hat{\tau}_n = \text{Enc}_t(\tau_n)$ and the video embedding $e_t$ in $M_t$ to retrieve task-relevant frames. Here, we use

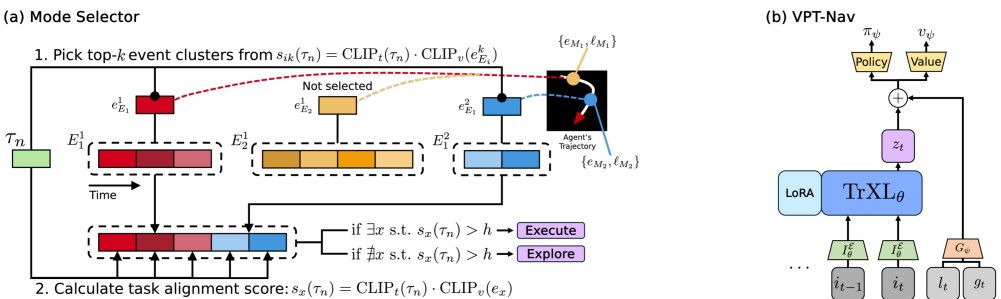

**Figure 3:** Mode Selector and VPT-Nav in MrSteve. (a) Mode Selector with Place Episodic Memory. It decides agent's mode (Explore or Execute) based on whether a task-relevant resource is in the memory. It uses a hierarchical read operation. (b) Architecture of Goal-Conditioned VPT Navigator.

video encoder $\text{Enc}_v$, text encoder $\text{Enc}_t$, and $H = 16$ from MineCLIP (Fan et al., 2022a), which is a CLIP (Radford et al., 2021) trained on web videos of Minecraft gameplay and associated captions.

While FIFO Memory offers benefits from its simple memory operations, it has two drawbacks. First, the computational complexity of the read operation grows linearly with the memory size. Second, the bias toward removing the oldest frames can be problematic in sparse sequential task as in the Figure 1 scenario, where task-relevant frames from visited places are lost.

**Place Memory** To address these issues, Place Memory (Cho et al., 2024) divides the agent's positions $\{\ell_n\}_{n=0}^{t}$ in the trajectory into clusters of distinct places, where each place cluster is assigned a FIFO Memory. Here, we term $\ell_t = (\text{coord}_x, \text{coord}_y, \text{yaw})$ as agent's position, which is a concatenation of agent's top-down location and its head direction. Place Memory is represented as $M_t = \{M_k\}_{k=1}^{K}$, where $M_k$ is the $k$-th place cluster with center position $\ell_{M_k}$, and center embedding $e_{M_k}$. Here, $e_{M_k}$ is the video embedding whose position is closest to $\ell_{M_k}$. This structure improves the efficiency of the read operation by extracting top-$k$ place clusters with their center embeddings first, then fetching relevant frames from these clusters. Furthermore, when memory capacity is limited, the oldest frame is removed from the largest place cluster, allowing the agent to retain memories in diverse places.

While Place Memory prioritizes storing experience frames across diverse places, its FIFO structure within each cluster still loses novel experience frames in the past. For instance, if an agent stays in a place where zombies burn and disappear for a long time, the place cluster removes the frames of burning zombies that can be crucial in upcoming tasks. This highlights the importance of focusing on visually distinct experience frames rather than storing them sequentially, which can be redundant.

**Place Event Memory** To resolve this issue, we introduce Place Event Memory built on Place Memory, which captures distinct events that occur within each place cluster (Figure 2(b)). While Place Memory uses agent's position to cluster experience frames, there is no criterion for clustering frames to form events. To tackle this, we use the cosine similarity of video embeddings from MineCLIP for criterion.

Specifically, each place cluster $M_k$ is subdivided into event clusters, denoted as $\{E_i^k\}_{i=1}^{d_k}$, where each $E_i^k$ represents the $i$-th event cluster in $k$-th place cluster, characterized by a center embedding $e_{E_i}^k$. These event clusters are newly created and updated as the place cluster accumulates a certain number of additional experience frames. For generating event clusters, DP-Means algorithm (Dinari & Freifeld, 2022) is applied on video embeddings of these frames, generated by MineCLIP, and the resulting cluster centers become a center embedding of each cluster. If the cosine similarity of the cluster centers between a newly created cluster and an existing cluster is higher than threshold $c$ (we define that two event clusters are indistinct), the two clusters are merged to prevent redundancy and ensure distinct event clusters within each place cluster. When memory capacity is exceeded, the oldest frame in the largest event cluster is removed, thus the memory can retain diverse places and distinct events within each place. More details on Place Event Memory can be found in Appendix E.

## 3.2 SOLVER MODULE: MODE SELECTOR, EXPLORATION, AND NAVIGATION

In this section, we introduce the remaining components in Solver Module, which are Mode Selector, and hierarchical policies $\pi_{\text{H-Cnt}}$, $\pi_{\text{L-Nav}}$ for episodic exploration, and goal-reaching navigation.

**Mode Selector** Mode Selector decides between Explore and Execute mode by checking whether task-relevant resource exists in the memory. If the resource exists, the agent chooses Execute mode, or Explore mode otherwise. When Place Event Memory is employed, Mode Selector first picks top-$k$ event clusters with task alignment score $s_{ik}(\tau_n) = \mathrm{CLIP}_t(\tau_n) \cdot \mathrm{CLIP}_v(e_{E_i}^k)$ between task embedding, and center embedding of event cluster. Then, it calculates task alignment scores on experience frames in top-$k$ event clusters and gathers frames with scores higher than task threshold $h$ as shown in Figure 3(a). We note that leveraging Place Event Memory offers computational efficiency with hierarchical read operation compared to FIFO Memory, which calculates the alignment scores on whole frames in the memory. We provide a comparison of memory query time in Appendix K for further insights.

**Hierarchical Episodic Exploration** We propose a memory-based hierarchical exploration method that allows the agent to efficiently explore the environment while minimizing revisits to previously explored positions. This is achieved through a high-level goal selector $\pi_{\mathrm{H\text{-}Nav}}(g_t | \ell'_t, M_t)$ and a low-level goal achiever $\pi_{\mathrm{L\text{-}Nav}}(a_t | h_t, g_t)$, where $h_t = i_{t-128:t}$, and $\ell'_t$, and $g_t$ are the agent's current location, and goal location in $(\mathrm{coord}_x, \mathrm{coord}_y)$, respectively. We introduce a **Count-Based** exploration strategy (Yamauchi, 1998; Tang et al., 2017; Chang et al., 2023) for the high-level goal selector.

Specifically, $L \times L$ visitation grid map $m_t$ is used with the agent's starting location set as the center of the map. The locations of the agent's trajectory are discretized and marked on the grid. The goal selector then divides the visitation map into grid cells of size $G \times G$, and selects the location of the grid cell with the lowest visitation count as the goal, $g_t$. If multiple grid cells have the same minimum count, the cell closest to the current location $\ell'_t$ is chosen. This approach directs the agent toward unexplored locations, while minimizing unnecessary revisits. The size of grid cell can be dynamically adjusted to balance between broader exploration and finer local searches. Additionally, in an infinitely large map, the visitation map can be easily expanded by adding new grids, and further hierarchies on visitation maps can be introduced for efficiently managing explored locations.

**Goal-Conditioned VPT Navigator** Once the goal location is selected by the high-level goal selector, it is crucial for the agent to navigate to the goal accurately. However, navigating complex terrains (*e.g.*, river, mountain) requires human prior knowledge, where pure RL policy trained from scratch in prior work (Yuan et al., 2023) often shows suboptimal navigation ability. To address this, we use the VPT as our starting policy, and fine-tune it for goal-conditioned navigation policy. We name this policy as **VPT-Nav**. In VPT-Nav, we add goal embedding $G_\psi(l_t, g_t)$ in the output of $\mathrm{TrXL}_\theta$ in VPT with LoRA adaptor (Hu et al., 2021a) as in Figure 3(b). We used PPO (Schulman et al., 2017) for fine-tuning goal encoder $G_\psi$, LoRA parameters, policy $\pi_\psi$, and value $v_\psi$ with reward based on the distance to the goal location. We note that our VPT-Nav introduces several differences from prior VPT fine-tuning methods, thoroughly investigated in Appendix L.

## 4 EXPERIMENTS

This section presents a step-by-step validation of our agent MrSteve across various environments and conditions. We begin by evaluating the exploration and navigation ability of MrSteve, which is crucial in sparse sequential tasks (Section 4.1). Then, we demonstrate MrSteve's capability to solve *A-B-A* task sequentially where the memory is necessary to solve the task *A* twice (Section 4.2). Additionally, we show that the proposed Place Event Memory outperforms other memory variants, particularly when memory capacity is limited (Section 4.3). Lastly, we showcase the generalization of MrSteve to long-horizon sparse sequential task (Section 4.4). Each baseline and task is explained in each of the experiment sections with more details in Appendix C.

### 4.1 EXPLORATION & NAVIGATION FOR SPARSE SEQUENTIAL TASK

In this section, we evaluate the exploration and navigation ability of our agent. To verify this, we placed an agent in a $100 \times 100$ block map with complex terrains such as mountains and a river and gave 6K steps to wander around the map. Since successful exploration in Minecraft involves covering as much of the map as possible while minimizing revisits to previously visited locations, we measure two metrics: Map Coverage and Revisit Count. Map Coverage is calculated by dividing the map into $11 \times 11$ grid cells and measuring the percentage of cells covered by the agent's trajectory. Revisit Count measures the average number of times the agent visits the same grid cell.

**Table 1:** Map Coverage and Revisit Count of different exploration policies. Our exploration method (High-Level: Count-Based, Low-Level: VPT-Nav) performs the best.

| High-Level | Count-based | | RNN-based | | Steve-1 |
|---|---|---|---|---|---|
| Low-Level | VPT-Nav (Ours) | DQN | VPT-Nav | DQN (Plan4MC) | |
| Map Coverage (↑) | **84.42 ± 0.06** | 31.83 ± 0.11 | 29.82 ± 0.07 | 16.36 ± 0.04 | 50.77 ± 0.13 |
| Revisit Count (↓) | **0.38 ± 0.6** | 4.47 ± 1.43 | 4.72 ± 1.73 | 6.2 ± 1.73 | 2.68 ± 1.36 |

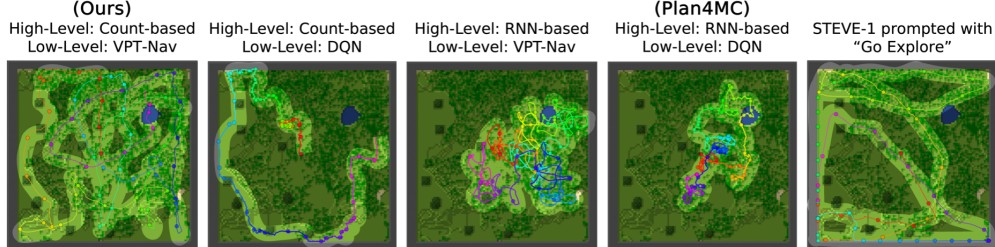

**Figure 4:** Agent's trajectories of length 6K steps on $100 \times 100$ block map with different exploration methods. The leftmost figure is the agent's trajectory from our exploration method.

To demonstrate that our proposed hierarchical episodic exploration with Count-Based high-level goal selector and low-level goal achiever VPT-Nav is more effective for exploration in the given map, we compared our method with the following baselines: Steve-1 (Lifshitz et al., 2024), and exploration method from Plan4MC (Yuan et al., 2023). For Steve-1, we provided the "Go Explore" instruction to assess its exploratory behavior as in prior works (Cai et al., 2023b; Zhou et al., 2024b). Plan4MC, on the other hand, employs a hierarchical approach where the high-level RNN policy selects the next goal location based on past locations, and a low-level DQN policy is used for goal-reaching. Since the input-output space of our method and Plan4MC is identical, interchanging high-level and low-level policies between two approaches is allowed so that we can evaluate the benefits of each component.

As shown in Table 1, and Figure 4, we can see that our exploration method outperforms other baselines. While Steve-1 showed decent performance in Map Coverage, it repeatedly visits previously explored places because of lack of memory. In the case of hierarchical exploration, high-level RNN policy struggled with memorizing visited places as the trajectory gets longer, resulting in high Revisit Counts. Additionally, the low-level DQN policy had difficulty navigating complex terrain, such as mountains and rivers, showing low Map Coverage. On the other hand, the Count-Based goal selector that directs the agent to the least-visited locations as goals and the VPT-Nav that effectively reaches those goals resulted in strong exploratory behavior. Furthermore, to show the robustness of VPT-Nav, we report its navigation capability in diverse terrains in Appendix L.

## 4.2 SEQUENTIAL TASK SOLVING WITH MEMORY IN SPARSE CONDITION

In this section, we demonstrate our agent's capability to solve sparse sequential tasks based on exploration methods studied in Section 4.1. To evaluate this, we introduce *ABA-Sparse* task, which consists of *A-B-A* tasks given sequentially with text instructions. Task *A* involves gathering a sparse resource, which can be either a water bucket, beef, wool, or milk. Task *B* requires collecting a dense resource, chosen from log, dirt, leaves, seeds, or sand, making total 20 tasks. The agent spawns in a $100 \times 100$ block map, and the *A* resource exists in a single location, while the *B* resource can be found in multiple locations. The agent is given 12K steps with unlimited memory capacity to complete all tasks. Since finding the sparse resource *A* is challenging, the task requires an efficient exploration algorithm. Moreover, after solving tasks *A* and *B*, memory becomes crucial to return to the location of resource *A* within the time limit. We measure success rates and task duration for evaluation.

To verify the benefits of efficient exploration and the memory, we compared the following agents: Steve-1, MrSteve with exploration method from Plan4MC and FIFO Memory (PMC-MrSteve-FM), and our agent MrSteve with Count-Based goal selector and VPT-Nav for exploration and Place Event Memory. We also test MrSteve with different memory variants, MrSteve-FM, MrSteve-EM,

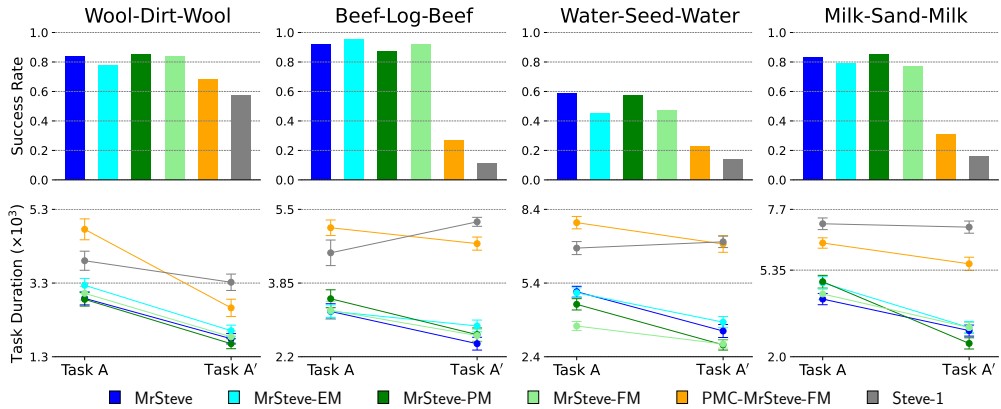

**Figure 5:** Success Rate and Task Duration of different agents in *ABA-Sparse* tasks. Task *A* refers to the first *A* task in the *A-B-A* task sequence, while Task *A′* refers to the final *A* task in the *A-B-A* task sequence. We note that MrSteve, as well as its memory variants, outperforms Steve-1, which lacks the memory. Additionally, while Steve-1 takes a similar amount of time to solve both task *A* and task *A′*, MrSteve solves task *A′* much faster. The full results on all 20 tasks are in Appendix H, and investigations about memory variants are in Appendix O.

MrSteve-PM, which use FIFO Memory, Event Memory, and Place Memory, respectively. Here, Event Memory is a Place Event Memory without place clusters, which stores the frames based on visual similarity. We explain the details of Event Memory in Appendix E.3.

As shown in Figure 5, it is clear that MrSteve outperforms Steve-1. This is because MrSteve can find task-relevant resources faster with efficient exploration and store the location of the task *A* resource, allowing it to revisit the location and solve task *A* again within a limited time. This is evident from the task duration in Figure 5, where MrSteve shows a shorter task duration than Steve-1 on the first *A* task. When solving the second *A* task, MrSteve exhibits a much shorter task duration compared to the first *A* task, while Steve-1 takes a similar or even greater number of steps. While other memory-augmented baselines showed similar performance to MrSteve, PMC-MrSteve-FM performed worse due to a suboptimal exploration method, making it difficult to find the sparse resource. We report the full results on all 20 tasks in Appendix H.

## 4.3 MEMORY-CONSTRAINED TASK SOLVING WITH MEMORY

We demonstrated in Section 4.2 that memory is essential in solving sparse sequential tasks when there is no limitation in memory capacity. However, in real-world scenarios where memory capacity is limited, memorizing visited places and novel events becomes important. In this section, we show that Place Event Memory can benefit in this scenario. To verify this, we introduce three Memory Tasks, which are *Find Water*, *Find Zombies' Death Spot*, and *Find First-Visited House*. In all tasks, the agent begins with an exploration phase, followed by a task phase. In the exploration phase, the agent follows a fixed exploratory behavior. In the subsequent task phase, the agent is given a MineCLIP embedding $e_t = \mathrm{Enc}_v(\{i_t\}_{n=1}^{16})$ as a task embedding instead of text instruction, where $i_t$ is the pixel observation seen in the exploration phase. Then, this becomes an image goal navigation task where an agent should navigate to the location of the given image.

In all tasks, the exploration phase is 3K steps, and the agent's memory capacity is limited to 2K. In *Find Water* task, the agent stays near water for $0.5$K steps, then travels to a random location. In the task phase, the agent should return to the water (Figure 6(a)). This task evaluates whether an agent can memorize water frames in the past. In *Find Zombies' Death Spot*, the agent sees burning zombies for 1K steps (zombies burn for $0.5$K steps then disappear), then travels. The task is to return to where zombies burned (Figure 6(b)). This task evaluates whether an agent can memorize distinct events (zombies burn and disappear) in the same place. In *Find First-Visited House* task, the agent sees the first house for $0.1$K steps, then goes to the second house and stays until 2K step, then travels. The task is to return to the first house (Figure 6(c)). This task evaluates whether an agent can memorize two visually similar houses in two different places.

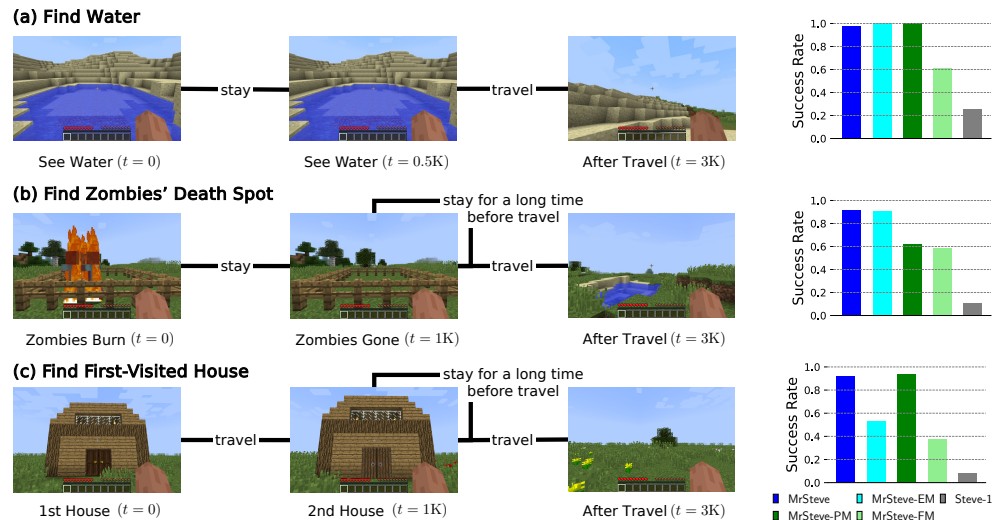

**Figure 6:** The overview of Memory Tasks, and Success Rate for each Memory Task from different agents. Memory Tasks are basically navigation tasks reaching the location of the previously seen experience frame. We observe that MrSteve which uses Place Event Memory shows high success rates in all tasks.

To evaluate how each memory type performs in the Memory Tasks, we tested Steve-1, MrSteve, MrSteve-FM, MrSteve-EM, and MrSteve-PM. In Figure 6, the performance of each memory type is illustrated. In all tasks, Steve-1 which lacks memory, showed worst performance since it has to find the targets from scratch. In *Find Water* task, MrSteve-FM does not have the water frames in memory in task phase, so it should explore to find the water showing about 60% success rate. In contrast, other agents store the water frames by allocating place cluster or event cluster in water location, allowing the agent to recall the frames and easily return to water, showing high success rate. In *Find Zombies' Death Spot* task, MrSteve-PM loses burning zombies' frames since Place Memory removes the frames in the largest place cluster, which is zombies' place, showing about 60% success rate. However, MrSteve-EM and MrSteve store the burning zombies' frames as a novel event, allowing the agent to easily return to zombies' spot, showing high success rate. In *Find First-Visited House* task, MrSteve-EM loses frames of first house, since it clusters two visually similar houses as the same event, showing about 50% success rate. However, MrSteve-PM and MrSteve store the two houses in different place clusters, enabling the agent to return to the first house, showing high success rate. These results suggest that Place Event Memory demonstrates its strength in memory-limited settings, where memorizing both visited places and novel events is crucial for task completion.

## 4.4 LONG-HORIZON SPARSE SEQUENTIAL TASK SOLVING WITH MEMORY

In this section, to see how MrSteve generalizes to long-horizon tasks, we introduce two sparse sequential tasks. For both tasks, the agent plays in a $200 \times 200$ block map for 500K steps (About 7 hours of gameplay) with 20K steps of memory capacity. The first is *Long-Instruction* task, where the agent is continuously given random tasks from *Obtain X*. Here $X$ can be water 🪣, beef 🥩, wool ⬜, log 🪵, dirt 🟫 or seeds 🌱. If the agent fails to complete the task within 20K steps, the task is changed. This task requires efficient exploration in a large map, and managing memory to memorize places with task-relevant resources to continuously solve the given tasks.

The second task is *Long-Navigation* task similar to Memory Tasks in Section 4.3. It has an exploration phase of 16K steps and a task phase. In the exploration phase, the agent observes six events in different places: 1) burning zombies, 2) river, 3) sugarcane blow up, 4) spider spawn, 5) tree, and 6) house, spending 2K steps at each place. In the task phase, the image goal is continuously given randomly selected from the frames in the early steps of the event. For instance, if the task is to reach sugarcane place, the image of sugarcane place before blow up is set as an image goal. This task requires managing memory to retain distinct events in different places.

The results are shown in Figure 7. For *Long-Instruction* task, we observe that MrSteve, and MrSteve-PM solved over 80 tasks, showing their capabilities of retaining task-relevant resources in different places effectively. MrSteve-EM solved around 50 tasks, suggesting event-based memory is less effective than place-based memory. This is because similar events in different places, like cows and sheeps living in visually similar forests, are in the same event cluster, possibly los-

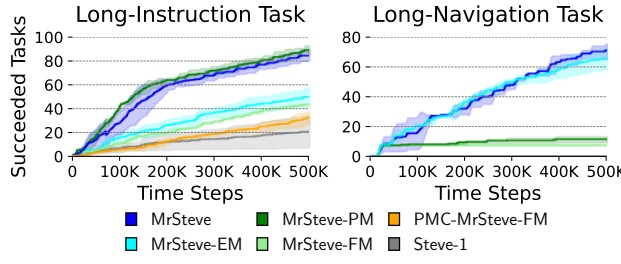

**Figure 7:** The performance in *Long-Intruction* task and *Long-Navigation* task. MrSteve performs well in both tasks.

ing task-relevant frames. For the remaining baselines, they either have suboptimal exploratory behaviors or keep losing the task-revelant frames, solving less than 50 tasks. For *Long-Navigation* task, MrSteve, and MrSteve-EM solved around 70 tasks, showing the ability to retain novel events occured in different places. In the case of MrSteve-PM, it removes the frames in early time steps of each place cluster, losing novel events (*e.g.*, sugarcane before blow up), thus solving less than 20 tasks. For the remaining baselines, they lose or cannot retain frames in the early stage of an episode solving less than 10 tasks. These results suggest that MrSteve demonstrates its strength in long-horizon tasks.

## 5 LIMITATIONS

This work focuses on improving Steve-1 through the introduction of What-Where-When episodic memory, significantly enhancing the agent's ability to retain and recall past events for more efficient task-solving. Our experiments demonstrate that MrSteve exhibits significantly enhanced performance when integrated with LLM-augmented agents for high-level planning. However, current limitations prevent the high-level planner from accessing PEM in the low-level controller. Future work could explore enabling PEM access for high-level planners, which could generate more accurate plans by leveraging the agent's episodic memories, further enhancing the system's capabilities for complex, long-horizon tasks.

We list up a few more limitations. First, our experiments are limited to surface-level exploration in the Minecraft environment, omitting underground navigation, which is a crucial aspect of the game as mining plays a central role. However, our hierarchical exploration methods based on visitation map could easily be extended to include vertical dimensions. Additionally, while our VPT-Nav demonstrated strong navigation abilities in plains biomes, more challenging terrains, such as crossing cliffs that require skills like building bridges, were not addressed. Lastly, we used exact position data in Minecraft, which may limit the model's adaptability to robotics tasks where positional information is often noisy. One possible direction is adapting MrSteve in environments with noisy positions.

## 6 CONCLUSION

In this paper, we introduced MrSteve (**M**emory **R**ecall Steve), a novel low-level controller designed to address the limitations of current LLM-augmented hierarchical approaches in general-purpose embodied AI environments like Minecraft. We argued that the primary cause of failures in many low-level controllers is the absence of an episodic memory system. To overcome this, we equipped MrSteve with Place Event Memory (PEM), a form of episodic memory that captures and organizes what, where, and when information from episodes. This allows for efficient recall and navigation in long-horizon tasks, directly addressing the limitations of existing low-level controllers like Steve-1, which rely heavily on short-term memory. Additionally, we proposed an Exploration Strategy and a Memory-Augmented Task Solving Framework, enabling agents to effectively switch between exploration and task-solving based on recalled events. Our results demonstrate significant improvements in both task-solving and exploration efficiency compared to existing methods. We believe that MrSteve opens new avenues for improving low-level controllers in hierarchical planning and are releasing our code to facilitate further research in this field.

## ETHICS STATEMENT

We acknowledge potential societal concerns related to our work. While our agent, in its current form, is designed for use in virtual environments such as Minecraft, the techniques and advancements made here could be extended to broader autonomous systems. There is a possibility that, if adapted for real-world applications, such systems might be misused for unethical purposes, including unauthorized surveillance or actions that infringe on individual privacy. We encourage responsible usage and further research into safeguards to prevent such outcomes.

## REPRODUCIBILITY STATEMENT

To facilitate the reproducibility of our work, we have provided the pseudo codes, model architecture, and hyperparameters in Appendix D, E, and F. We will also release the source code for our models and experiments.

## ACKNOWLEDGEMENT

This work was supported by GRDC (Global Research Development Center) Cooperative Hub Program (RS-2024-00436165), Brain Pool Plus Program (No. 2021H1D3A2A03103645), and Basic Research Lab (No. RS-2024-00414822) through the National Research Foundation (NRF) funded by the Ministry of Science and ICT (MSIT). The authors would like to thank the members of Machine Learning and Mind Lab (MLML) for helpful comments.

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

## A    LIMITATIONS OF MEMORY SYSTEM IN LLM-AUGMENTED AGENTS

Memory systems in agents primarily aim to retrieve robust and accurate high-level plans for long-horizon tasks (Wang et al., 2023a;b; Zhu et al., 2023; Qin et al., 2024; Li et al., 2024). Existing works use an abstracted memory that stores the succeeded task with its plans, and often with history of observations for reliable retrieval, which is helpful when the similar plan in other tasks already exists in memory. However, these types of memory systems are not well-suited for low-level controllers for the following reasons:

- **Issues with Managing Memory** Recent memory systems in Minecraft, when saving successful plans or skills, the history of observations for solving the task is all stored in FIFO manner (Wang et al., 2023b) or only task-relevant frames are stored in the plan (Li et al., 2024). However, computation complexity for retrieving the experience frames that are relevant to a new task is computationally expensive or even impossible (because the latter may not store the experience frames despite the agent observing it).

- **Lack of Mechanism for Retrieving Experience frames** Current memory systems store the text instructions of succeeded tasks as keys and their plans to complete those tasks as values. The retrieval process begins by matching a task query to the task keys in memory and then filtering further using the current scene's similarity to the stored frames before retrieving the final plan. These memory systems are targeted to retrieve the succeeded plan, but they lack a mechanism for utilizing the experience frames in the memory, which could be crucial for future tasks.

Consider the following example: Suppose the agent is tasked with collecting wood. While searching for a tree, let's assume the agent came across a cow in the forest. Once the wood is collected, the memory will store the successful plan and its corresponding observations. However, if the agent is later tasked with finding the cow, there would be no memory key related to the cow, making it impossible to retrieve the relevant frames. While some heuristics to calculate the similarity of the task embedding for "find cow" and the visual representations in memory using MineCLIP is possible, it is computationally expensive since the similarity should be calculated for all stored frames.

Thus, we need a new type of memory system for the low-level controller to efficiently store novel events (such as encountering the cow) as they explore the environment, even when such events are not directly relevant to the current task. Also, the memory should hierarchically organize these novel events so that they can be efficiently retrieved later. In this paper, we propose a memory system called Place Event Memory (PEM), which organizes experiences by both location and event. PEM allows the agent to store diverse novel events across various locations, making future retrieval more efficient. We argue that PEM, when combined with current memory systems that store successful plans, will enable more effective retrieval of task-relevant information.

## B    COMPUTATION OVERHEAD

Our study was performed on an Intel server equipped with 8 NVIDIA RTX 4090 GPUs and 512GB of memory. The inference time for tasks under 20K steps for running a single episode was approximately 30 minutes on a single GPU. For long-horizon tasks that take 500K steps, approximately 12 hours were required for running a single episode on a single GPU. The VPT-Nav training took roughly 23 hours on a single GPU.

## C    ENVIRONMENTS DETAILS

### C.1    ENVIRONMENTS SETTING

All tasks are implemented using MineDojo (Fan et al., 2022b). We utilize MineDojo's success checker, where the success of each task is determined based on changes in the agent's inventory. Hence, the agent succeeds in the task if the corresponding target item appears in the inventory. If the agent exceeds the time limit of each task or dies before completing assigned tasks, indicating the agent failed the task.

For all tasks, we assume that the agent has access to both first-person view pixels and its positional data. The raw pixel observation $i_t$ is provided in the shape $(160 \times 256 \times 3)$. The agent's position $p_t$

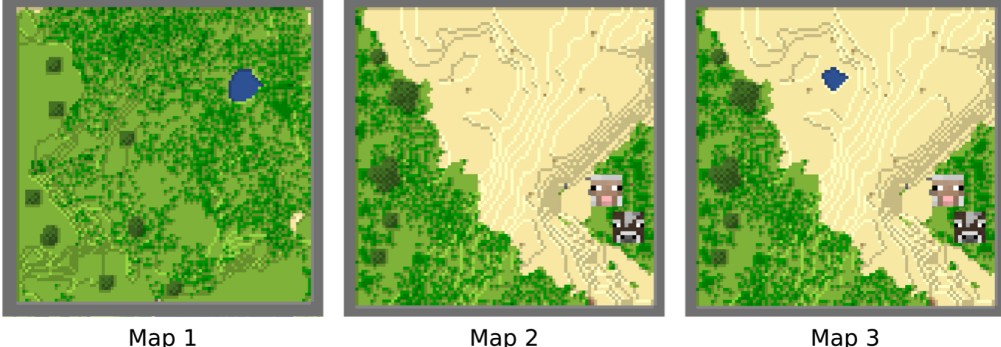

Map 1          Map 2          Map 3

**Figure 8:** Topdown View of Three *ABA-Sparse* Task Maps. The first map was used in the *ABA-Sparse* tasks, including the water bucket task. Trees are distributed on the left side of the map, and water exists only in the upper right corner. The second map was used in tasks, not including the water bucket and sand task. Trees are located on the left side of the map, and on the right side, there are cows and sheep, with a mountain separating them. The last map was used in tasks, including the sand task. Its overall layout is identical to the second map, except for an additional water pond at the top. The map's edges are surrounded by high walls, making it impossible to access anything other than the resources visible in the top-down view.

is represented as a vector of shape $(5, )$, where the first three components correspond to the $(x, y, z)$ coordinates, and the last two components represent the pitch and yaw angles of the agent's camera. We note that no privileged observations, such as LiDAR or voxel data, are provided. The agent operates in a keyboard and mouse action space following VPT (Baker et al., 2022). This action space consists of button input states paired with mouse movements.

Crafting items, which requires long-horizon planning, is not considered in our method. To eliminate the need for crafting, the appropriate item necessary for solving a task is provided to the agent at the beginning of each new task. For instance, if the task is to "obtain water bucket 🪣," the agent starts with an empty bucket 🪣 in its main hand. Additionally, we apply the following rules:

- `/difficulty peaceful`: This rule prevents the occurrence of hostile mobs, such as zombies and spiders, and death by starvation.

- `/gamerule doWeatherCycle false`: This rule keeps the weather clear to reduce the noise from heavy rain.

## C.2 TASK DETAILS

In this section, we describe the details for each task in Experiment section (Section 4). The basic environment settings follow those outlined in Appendix C.1, unless specified otherwise.

### C.2.1 EVALUATION PROTOCOLS

Except for the *Long-Horizon* tasks in Section 4.4, we ran 100 episodes for each agent using different random seeds to evaluate performance. For the *Long-Horizon* tasks, we reported the average success rate with the standard error over 5 episode runs for each agent.

### C.2.2 EXPLORATION & NAVIGATION TASK DETAILS

For this task, we used Map 1 in Figure 8 with $100 \times 100$ size, surrounded by high walls. The map includes complex terrains such as mountains and a water pond, which requires a robust low-level navigation policy for successful exploration. For each episode, the agent spawns in the center of the map and explores for 6,000 steps.

### C.2.3 *ABA-Sparse* TASKS DETAILS

In these tasks, the agent is asked to complete three sequentially given tasks, where the first and third tasks are identical. The target item for the first task, denoted as $A$, is one of four sparsely distributed items: water 🪣, beef 🥩, wool 🧶, or milk 🥛. In the second task, denoted as $B$, the taget item is one of five items: log 🪵, dirt 🟫, seeds 🌱, leaves 🌿, or sand 🟨. The agent has unlimited memory capacity and is allowed a maximum of 12,000 steps to complete three tasks. The task only changes to the next one upon successful completion of the current task.

If the first task, $A$, is to obtain a water bucket 🪣, the agents spawn at the center of Map 1, shown in Figure 8. On this map, most of the surface is covered with dirt, while grass, which provides seeds, is widely distributed. A water pond is located in the upper right corner of the map, but it only becomes visible when agents are nearby.

If the first task, $A$, is to collect beef 🥩 or wool 🧶, the agents spawn at the center of Map 2 in Figure 8. Similarly, if the second task, $B$ is to collect sand 🟨, the agents start at the center of Map 3 in Figure 8. In both maps, trees are scattered on the left side of the map, while sheep 🐑 and cows 🐮 are found on the right side. A sand mountain runs through the middle, separating the trees from the sheep and cows. Dirt is only present on the far left and right sides of the map.

### C.2.4 MEMORY TASKS

In the three Memory Tasks, the agent explores for 3,000 steps before being asked to complete an image goal navigation. Unlike the *ABA-Sparse* Tasks, the agent has a limited memory capacity of 2,000 frames. Consequently, an agent utilizing the FIFO memory forgets the memory from the first 1,000 frames after the exploration phase.

**Find Water Task** The agent is initially spawned near a water pond and remains in its vicinity for 500 steps. During this period, one observation is selected as a goal image for a subsequent navigation task. After the initial 500 steps, the agent moves to a random location for the remainder of the exploration phase. After 3K steps, the agent begins to navigate back to the water pond it observed at the start of the episode.

**Find Zombies' Death Spot Task** At the beginning, the agent sees burning zombies for the first 1K steps. Approximately 500 steps after the start of the episode, the zombies disappear, resulting in the agent observing two distinct scenes in the same location. A goal image for navigation is selected from the observation where the zombies are burning. After 1K steps, the agent starts to travel for the rest of the exploration phase. Once the exploration phase finishes, the agent returns to the place where the zombies were burning.

**Find First-Visited House Task** In this task, there are two distinct houses that look similar to one another. The agent starts near one of the houses, where it stays for 100 steps before moving to the other house, where it remains for 2K steps. For the remainder of the exploration phase, the agent travels to a random location. After the exploration phase, the agent is asked to go to the first house it has visited.

### C.2.5 LONG-HORIZON TASKS

**Long-Instruction Task** In this task, the agent is required to complete a series of tasks sequentially on a $200 \times 200$ block-sized map in Figure 9. The order of tasks within the sequence is randomized, with each task being one of six possible types: water bucket 🪣, beef 🥩, wool 🧶, wood 🪵, dirt 🟫, and seeds 🌱. We evaluated the agent's performance by measuring the number of successfully completed tasks over 500K steps. If the agent fails to complete a given task within 20,000 steps, that task is canceled, and a new one is assigned.

**Long-Navigation Task** This task is basically image goal navigation in a $200 \times 200$ block sized map in Figure 9. Before the task phase, the agent undergoes a 16K-step exploration phase where it observes six landmarks: 1) zombie burning 🧟, 2) water 🪣, 3) sugarcane explosion 🎋, 4) spider spawn 🕷, 5) tree 🌲, and 6) house, spending 2K steps at each location. In the task phase, the agent is given a random start image of one of these landmarks and must navigate to it. The dynamic nature of some landmarks, such as the burning zombie, exploding sugarcane, and spider spawn makes it important to store novel events from the exploration phase.

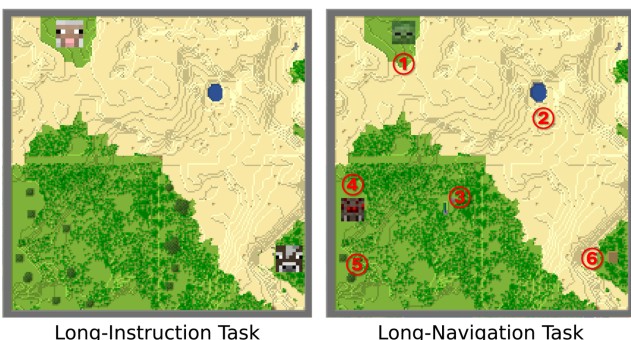

Long-Instruction Task          Long-Navigation Task

**Figure 9:** Topdown View of Two *Long-Horizon* Task Maps. Both maps are of size $200 \times 200$ blocks. (Left) This map is used for the *Long-Instruction* task. Trees are located in the lower left and middle bottom of the map, while sheep inhabit the upper left area, and cows exist in the right bottom. A water pond can be found in the upper right area of the map. (Right) This map is utilized for the *Long-Navigation* task. Agents traverse between six scenes in the map. Three of them are dynamic scenes: burning zombies, popping sugarcanes, and spawning spiders, which are positioned at the first, third, and fourth places on the map, respectively. The other three are static scenes: a water pond, trees, and a house, located at the second, fifth, and sixth places on the map, respectively.

# D MRSTEVE ALGORITHM

---

**Algorithm 2** MrSteve Algorithm

---

**Require:** $X_{t_n^{\text{init}}}$ from Environment env, Memory $M_{t_n^{\text{init}}}$, and new task $\tau_n$ at time step $t_n^{\text{init}}$.
 1: mode ← null
 2: $T_{\text{mode}} \leftarrow 0$
 3: $g_t \leftarrow$ null
 4: reached ← false
 5: $t \leftarrow t_n^{\text{init}}$
 6: $X_t \leftarrow X_{t_n^{\text{init}}}; M_t \leftarrow M_{t_n^{\text{init}}}$
 7: **loop**
 8:     Write$(M_t, X_t)$
 9:     **if** $T_{\text{mode}} > L$ **then**
10:         mode ← null
11:         $T_{\text{mode}} \leftarrow 0$
12:         $g_t \leftarrow$ null
13:         reached ← false
14:     **end if**
15:     **# Mode Selector**
16:     **if** mode is null **then**
17:         candidates ← Read$(M_t, \tau_n)$
18:         **if** candidates $\neq \emptyset$ **then**
19:             $X_t' \leftarrow$ PickOne(candidates)
20:             $g_t \leftarrow$ Position$(X_t')$
21:             mode ← EXECUTE
22:         **else**
23:             mode ← EXPLORE
24:         **end if**
25:     **end if**
26:     $h_t \leftarrow X_{t-128:t}$
27:     **# Explore Mode**
28:     **if** mode is EXPLORE **then**
29:         $g_t \sim \pi_{\text{H-Nav}}(g_t|\ell_t', M_t)$
30:         $a_t \sim \pi_{\text{L-Nav}}(a_t|h_t, g_t)$
31:     **end if**
32:     **# Execute Mode**
33:     **if** mode is EXECUTE **then**
34:         **if** the agent reaches $g_t$ **then**
35:             reached ← true
36:         **end if**
37:         **if** reached **then**
38:             $a_t \sim \pi_{\text{Inst}}(a_t|h_t, \tau_n)$
39:         **else**
40:             $a_t \sim \pi_{\text{L-Nav}}(a_t|h_t, g_t)$
41:             $g_{t+1} \leftarrow g_t$
42:         **end if**
43:     **end if**
44:     **if** the task $\tau_n$ succeeded or timeout **then**
45:         **break**
46:     **end if**
47:     $T_{\text{mode}} \leftarrow T_{\text{mode}} + 1$
48:     $X_{t+1} \leftarrow$ env.step$(a_t); M_{t+1} \leftarrow M_t$
49:     $t \leftarrow t + 1$
50: **end loop**

---

We provide more details in Algorithm 2. This algorithm is executed when a new task is given to the MrSteve and finished if it completes the task or exceeds a time limit (Line 44-46).

**Agent State Variables** When a new task $\tau_n$ is given, the agent state variables are initialized (Line 1-4). `mode` indicates which module is executed and $T_{\text{mode}}$ denotes elapsed times for the current mode execution. $g_t$ is a target location for the navigation, and `reached` indicates whether the agent has reached the target location in the Execute Mode.

The Agent State Variables are reset if the agent remains in the current mode for $L$ steps (Lines 9-13). If the Mode Selector retrieves a wrong experience frame, where no task-relevant resource is present at the corresponding location, the agent can have trouble and fail to complete the given task for a long time. In addition, the agent might encounter previously unobserved task-relevant resources while navigating to the location of the previously retrieved memory. In both scenarios, changing the navigation target can be helpful. Consequently, the agent queries the memory to search for new candidates if the agent executes the current mode for $L = 600$ steps, equivalent to 30 seconds in-game time.

**Memory Write Frequency** In Line 8, MrSteve writes the current observation to the memory, regardless of the current mode.

**Mode Selector** The Mode Selector decides between two modes according to the existence of task-relevant resources in the memory. First, if no mode has been selected, MrSteve queries the memory (Line 16-17). If a task-relevant resource exists, MrSteve picks one of them and sets the navigation target $g_t$ to the location of the picked one (Line 18-21). Otherwise, the Explore Mode is selected (Line 23).

**Explore Mode** In the Explore Mode, MrSteve explores the least-visited locations using $\pi_{\text{H-Nav}}$ and $\pi_{\text{L-Nav}}$ (Line 28-31). We provide more details in Appendix F.2.

**Execute Mode** In the Execute Mode, MrSteve navigates to the selected target location $g_t$ first. Once it reached, Steve-1 is executed to follow the task instruction $\tau_n$ (Line 33-43).

# E  PLACE EVENT MEMORY, PLACE MEMORY, EVENT MEMORY DETAILS

We provide the Algorithms on Write & Read operations of Place Event Memory and other memory variants, which are Place Memory and Event Memory. We also provide the specifications of each memory in the following section E.4.

## E.1  PLACE EVENT MEMORY WRITE & READ OPERATIONS

---

**Algorithm 3** Place Event Memory Write Operation at time step $t$

---

**Require:** Assume memory at time step $t$ as $M_t = \{M_k\}_{k=1}^K$ where $M_k$ is $k$-th place cluster with center position $\ell_{M_k}$, and center embedding $e_{M_k}$, and each place cluster $M_k$ has event clusters $\{E_i^k\}_{i=1}^{d_k}$ where $E_i^k$ is $i$-th event cluster with center embedding $e_{E_i}^k$. Each place cluster has dummy deque $Q_k$ that stores recent $R$ frames in that cluster, and update frequency timer $r_k$. Additional variables are Memory Capacity $N$, MineCLIP image encoder $\text{CLIP}_v$, and experience $X_t = \{i_t, l_t, t\}$ at time step $t$. MineCLIP threshold $c$.

1: **# Memory Add**
2: Get experience frame $x_t = \{e_t, l_t, t\}$ where $e_t = \text{CLIP}_v(i_t)$
3: **if** $\ell_t \in \text{PLACE\_CLUSTER}(M_t)$ **then**
4:     Find place cluster $M_j$ where $\text{PLACE}(\ell_t) = \ell_{M_j}$
5:     Add $x_t$ to dummy deque $Q_j$
6:     Update frequency timer $r_k = r_k + 1$
7:     **if** $r_k = R$ **then**
8:         $\{E_i, e_i\}_{i=1}^{l'} = \text{DP-Means}(Q_j)$
9:         $\{E_i, e_i\}_{i=1}^{l} = \text{MERGE\_CLUSTERS}(\{E_i, e_i\}_{i=1}^{l'})$
10:         **for** $u = 1, \ldots, l$ **do**
11:             add_cluster=True
12:             **for** $p = 1, \ldots, d_j$ **do**
13:                 **if** $e_u^j \cdot e_{E_p}^j > c$ **then**
14:                     Merge $E_u$ to $E_p^j$
15:                     add_cluster=False; break
16:                 **end if**
17:             **end for**
18:             **if** add_cluster=True **then**
19:                 Create new event cluster $E_{d_j+1}^j = E_u$ with center embedding $e_{E_{d_j+1}}^j = e_u$
20:                 Add created cluster $E_{d_j+1}^j$ to $M_j$
21:             **end if**
22:         **end for**
23:         $r_k = 0$
24:     **end if**
25: **else**
26:     Create new place cluster $M_{K+1}$ with center position $p_{M_{K+1}} = \text{PLACE}(\ell_t)$, and center embedding $e_{M_{K+1}} = e_t$
27:     Create new dummy deque $Q_{K+1} = \{x_t\}$
28:     Add created cluster $M_{K+1} = \{Q_{K+1}\}$ to $M_t$
29: **end if**
30: **# Memory Remove**
31: **if** $\text{len}(M_t) > N$ **then**
32:     Find event cluster $E_i^k$ where $i, k = \arg\max \text{len}(E_i^k)$
33:     Remove the oldest frame in $E_i^k$
34: **end if**

---

We further elaborate on how **# Memory Add** in Algorithm 3 operates. First, we get the experience frame $x_t$, and check if $\ell_t$ belongs to one of place clusters in $M_t$ (Line 3). Here, PLACE_CLUSTER() indicates the whole 2D space that memory $M_t$ covers. Since we use fixed-size square area for each place cluster, it can be seen as covered area of set of squares. If $x_t$ is not in current place clusters, new place cluster is created with dummy deque (Line 26-28). Dummy deque here is short memory

for each place cluster used for clustering the events. If $x_t$ belongs to some place cluster (Line 4, PLACE() maps $\ell_t$ to its place), it is added to dummy deque in the place cluster, and increase update frequency timer in that place cluster by 1 (Line 4-6). When update frequency timer equals to $R$, DP-Means algorithm (Dinari & Freifeld, 2022) is applied to experience frames in dummy deque for event clustering. After applying DP-Means algorithm, we get a set of events and its center embeddings (Line 8). However, we found that DP-Means tends to make different clusters even when the agent observes the same scenes. Thus, we applied MERGE_CLUSTERS() to DP-Means output clusters to merge clusters that have high alignments in center embeddings (Line 9). After this, for each newly created event cluster from DP-Means (Line 10), if some existing event cluster is aligned (Line 13), two clusters are merged (Line 14). If newly created cluster does not belong to any existing event clusters, then add it to the place cluster as a new event cluster (Line 19-20).

**Event Cluster Details** We use the DP-Means algorithm for clustering the events. DP-Means algorithm is a Bayesian non-parametric extension of the K-means algorithm based on small variance asymptotic approximation of the Dirichlet Process Mixture Model. It doesn't require prior knowledge on the number of clusters $K$. To run this algorithm, we first set the initial number of clusters $K'$ and cluster the data with K-Means++ initialization ($K'$ can be 1), then DP-Means algorithm automatically re-adjust the number of clusters based on the data points and cluster penalty parameter $\delta$. Thus, DP-Means algorithm behaves similarly to K-means with the exception that a new cluster is formed whenever a data point is farther than $\delta$ away from every existing cluster centroid.

Using clustering algorithm in Minecraft was previously done in Yuan et al. (2024), where K-Means algorithm is used to cluster 100 goal states in massive Minecraft dataset. However, they applied the clustering on the reduced dimension of MineCLIP representation with t-SNE (van der Maaten & Hinton, 2008). In this work, we directly apply DP-Means algorithm in MineCLIP representation space setting initial number of clusters $K' = 5$, and $\delta = 1$.

**Place Cluster Details** We provide a detailed explanation of how place clusters are formed. Each place cluster stores the agent's experience frames based on its 2D ground position $(x, y)$ and head direction angle yaw. The size of a place cluster and its yaw range are defined by parameters $C$ (in Minecraft block) and $W \in (0°, 180°)$, respectively, which distinguish different clusters. A place cluster is centered at a specific position $(x, y)$ with a center yaw $w$. An experience frame is assigned to a place cluster if the agent's position falls within the range $(x - C/2, x + C/2)$ for $x$ and $(y - C/2, y + C/2)$ for $y$, and the yaw angle satisfies $(w - W/2, w + W/2)$. We use relative position and yaw for center of the place cluster, implying that the center of the initial place cluster has position $(0, 0)$, and yaw 0.

---

**Algorithm 4** Place Event Memory Read Operation at time step $t$

---

**Require:** Assume memory at time step $t$ as $M_t = \{M_k\}_{k=1}^K$ where $M_k$ is $k$-th place cluster with center position $\ell_{M_k}$, and center embedding $e_{M_k}$, and each place cluster $M_k$ has event clusters $\{E_i^k\}_{i=1}^{d_k}$ where $E_i^k$ is $i$-th event cluster with center embedding $e_{E_i}^k$. Additional variables are MineCLIP image encoder $\text{CLIP}_v$, and text encoder $\text{CLIP}_t$, task instruction $\tau_n$ at time step $t$, and task threshold $h$.
1: Calculate $s_{ik}(\tau_n) = \text{CLIP}_t(\tau_n) \cdot \text{CLIP}_v(e_{E_i}^k)$ for all $i, k$
2: Sort $s_{ik}(\tau_n)$ for all event clusters and select top-K event clusters
3: Gather all experience frames $x_t = \{e_t, l_t, t\}$ from the top-K event clusters
4: cand = {}
5: **for** each experience frame $x_t$ in the selected top-K event clusters **do**
6:     $s_{x_t}(\tau_n) = \text{CLIP}_t(\tau_n) \cdot \text{CLIP}_v(e_t)$
7:     **if** $s_{x_t}(\tau_n) > h$ **then**
8:         Add $x_t$ to cand
9:     **end if**
10: **end for**
11: **return** cand

---

**Exploiting hierarchical structure of Place Event Memory** For memory read operation, we can exploit hierarchical structure of place event memory for better computation efficiency. Suppose we have $N_1$ place clusters and exactly $N_2$ event clusters for each place cluster. Then, read operation in Algorithm 4 has computational complexity of $O(N_1 N_2)$. However, if we first attend only to place clusters (calculate $s_{ik}$ from center embeddings $e_{M_k}$ from place clusters), then attend to event

clusters from extracted top-$k$ place clusters, the computational complexity becomes $O(N_1 + kN_2)$. However, center embedding $e_{M_k}$ may not be sufficient summarization of the place cluster, since center embedding can not capture all different events occurred in that place. Thus, we use the read operation as in Algorithm 4.

### E.2 PLACE MEMORY WRITE & READ OPERATIONS

---

**Algorithm 5** Place Memory Write Operation at time step $t$

---

**Require:** Assume memory at time step $t$ as $M_t = \{M_k\}_{k=1}^K$ where $M_k$ is $k$-th place cluster with center position $\ell_{M_k}$, and center embedding $e_{M_k}$. Each place cluster $M_k$ is a FIFO Memory. Additional variables are Memory Capacity $N$, MineCLIP image encoder $\text{CLIP}_v$, and experience $X_t = \{i_t, l_t, t\}$ at time step $t$.
1: **# Memory Add**
2: Get experience frame $x_t = \{e_t, l_t, t\}$ where $e_t = \text{CLIP}_v(i_t)$
3: **if** $\ell_t \in \text{PLACE\_CLUSTER}(M_t)$ **then**
4:     Find place cluster $M_j$ where $\text{PLACE}(e_t) = e_{M_j}$
5:     Add $x_t$ to $M_j$
6: **else**
7:     Create new place cluster $M_{K+1}$ with center position $\ell_{M_{K+1}} = \text{PLACE}(\ell_t)$, and center embedding $e_{M_{K+1}} = e_t$
8:     Add created cluster $M_{K+1} = \{x_t\}$ to $M_t$
9: **end if**
10: **# Memory Remove**
11: **if** $\text{len}(M_t) > N$ **then**
12:     Find place cluster $M_k$ where $k = \arg\max \text{len}(M_k)$
13:     Remove the oldest frame in $M_k$
14: **end if**

---

---

**Algorithm 6** Place Memory Read Operation at time step $t$

---

**Require:** Assume memory at time step $t$ as $M_t = \{M_k\}_{k=1}^K$ where $M_k$ is $k$-th place cluster with center position $\ell_{M_k}$, and center embedding $e_{M_k}$. Each place cluster $M_k$ is a FIFO Memory. Additional variables are MineCLIP image encoder $\text{CLIP}_v$, and text encoder $\text{CLIP}_t$, task instruction $\tau_n$ at time step $t$, and task threshold $h$.
1: Calculate $s_k(\tau_n) = \text{CLIP}_t(\tau_n) \cdot \text{CLIP}_v(e_{M_k})$ for all $k$
2: Sort $s_k(\tau_n)$ for all place clusters and select top-K place clusters
3: Gather all experience frames $x_t = \{e_t, l_t, t\}$ from the top-K place clusters
4: cand = {}
5: **for** each experience frame $x_t$ in the selected top-K place clusters **do**
6:     $s_{x_t}(\tau_n) = \text{CLIP}_t(\tau_n) \cdot \text{CLIP}_v(e_t)$
7:     **if** $s_{x_t}(\tau_n) > h$ **then**
8:         Add $x_t$ to cand
9:     **end if**
10: **end for**
11: **return** cand

---

## E.3 EVENT MEMORY WRITE & READ OPERATIONS

---

**Algorithm 7** Event Memory Write Operation at time step $t$

---

**Require:** Assume memory at time step $t$ as $M_t = \{E_k\}_{k=1}^K$ where $E_k$ is $k$-th event cluster with center embedding $e_{E_k}$, and each event cluster is a FIFO Memory. Memory has a dummy deque $Q$ that stores recent $R$ frames, and update frequency timer $r$. Additional variables are Memory Capacity $N$, MineCLIP image encoder $\text{CLIP}_v$, and experience $X_t = \{i_t, l_t, t\}$ at time step $t$. MineCLIP threshold $c$.

1: **# Memory Add**
2: Get experience frame $x_t = \{e_t, l_t, t\}$ where $e_t = \text{CLIP}_v(i_t)$
3: Add $x_t$ to dummy deque $Q$
4: Update frequency timer $r = r + 1$
5: **if** $r = R$ **then**
6:     $\{E_i, e_i\}_{i=1}^{l'} = \text{DP-Means}(Q_j)$
7:     $\{E_i, e_i\}_{i=1}^{l} = \text{MERGE\_CLUSTERS}(\{E_i, e_i\}_{i=1}^{l'})$
8:     **for** $u = 1, \ldots, l$ **do**
9:         add_cluster=True
10:         **for** $p = 1, \ldots, k$ **do**
11:             **if** $e_u^j \cdot e_{E_p} > c$ **then**
12:                 Merge $E_u$ to $E_p$
13:                 add_cluster=False; break
14:             **end if**
15:         **end for**
16:         **if** add_cluster=True **then**
17:             Create new event cluster $E_{k+1} = E_u$ with center embedding $e_{E_{k+1}} = e_u$
18:             Add created cluster $E_{k+1}$ to $M_t$
19:         **end if**
20:     **end for**
21:     $r_k = 0$
22: **end if**
23: **# Memory Remove**
24: **if** $\text{len}(M_t) > N$ **then**
25:     Find event cluster $E_k$ where $k = \arg\max \text{len}(E_k)$
26:     Remove the oldest frame in $E_k$
27: **end if**

---

**Algorithm 8** Event Memory Read Operation at time step $t$

---

**Require:** Assume memory at time step $t$ as $M_t = \{E_k\}_{k=1}^K$ where $E_k$ is $k$-th event cluster with center embedding $e_{E_k}$, and each event cluster $E_k$ is a FIFO Memory. Additional variables are MineCLIP image encoder $\text{CLIP}_v$, and text encoder $\text{CLIP}_t$, task instruction $\tau_n$ at time step $t$, and task threshold $h$.

1: Calculate $s_k(\tau_n) = \text{CLIP}_t(\tau_n) \cdot \text{CLIP}_v(e_{E_k})$ for all $k$
2: Sort $s_k(\tau_n)$ for all event clusters and select top-K event clusters
3: Gather all experience frames $x_t = \{e_t, l_t, t\}$ from the top-K event clusters
4: cand = {}
5: **for** each experience frame $x_t$ in the selected top-K event clusters **do**
6:     $s_{x_t}(\tau_n) = \text{CLIP}_t(\tau_n) \cdot \text{CLIP}_v(e_t)$
7:     **if** $s_{x_t}(\tau_n) > h$ **then**
8:         Add $x_t$ to cand
9:     **end if**
10: **end for**
11: **return** cand

---

**Discussions on MineCLIP threshold** As in Event Memory's Write Operation in Algorithm 7, clusters generated by DP-Means algorithm are either merged with an existing event cluster, or added as new event cluster. This is determined by MineCLIP threshold $c$, which is the criterion for separating

event clusters. We note that using proper value for threshold is important for Event Memory to work reasonably across the tasks. If $c$ is too small, Event Memory will cluster experience frames with only a few clusters, which may not be visually distinctive. If $c$ is too large, in extreme case, Event Memory will make event cluster for each experience frame. When memory capacity is exceeded, it will randomly remove the oldest frame (since all event clusters have the same size), which behaves as a FIFO Memory.

**Drawback in Event Memory** Event Memory removes the frame in the largest event cluster, when memory capacity is exceeded. This indicates that the memory can retain visually distinct events. However, since it does not consider position information in clustering, similar visual frames in different places can be clustered into the same event cluster. This can be fatal since task-relevant frames in different places can be removed from the memory, which can be crucial in upcoming tasks. This drawback is shown in *Find First-Visited House* task in Section 4.3, and *Long-Instruction* task in Section 4.4, where MrSteve-EM showed suboptimal performances.

### E.4  HYPER-PARAMETERS FOR MEMORY

We provide the specifications for each memory type used in the experiments. The task threshold $h$ is set to $22.74$ by default, but it can be adjusted to enhance retrieval accuracy.

**Table 2:** Specifications for each memory type.

| Memory Type | Parameter | Value |
|---|---|---|
| Place Event Memory | Place Cluster Size $C$ | 6 |
| | Place Cluster Yaw $Y$ | 60 |
| | Update Frequency $r_k$ | 100 |
| | top-$K$ | 30 |
| | MineCLIP Threshold $c$ | 73.5 |
| | Task Threshold $h$ | 22.74 |
| Event Memory | Update Frequency $r_k$ | 100 |
| | top-$K$ | 30 |
| | MineCLIP Threshold $c$ | 73.5 |
| | Task Threshold $h$ | 22.74 |
| Place Memory | Place Cluster Size $C$ | 6 |
| | Place Cluster Yaw $Y$ | 60 |
| | top-$K$ | 30 |
| | Task Threshold $h$ | 22.74 |

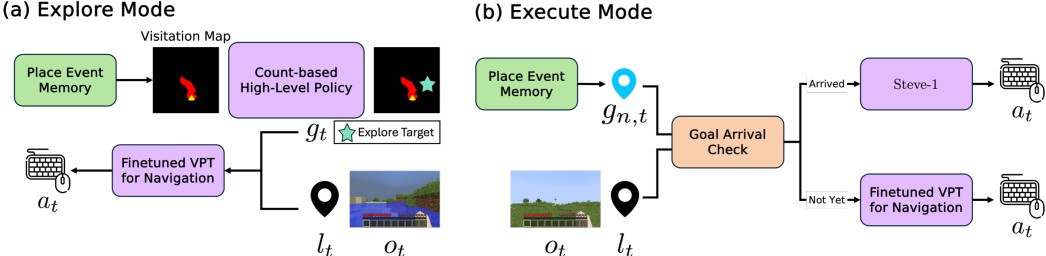

**Figure 10:** The model architectures of Explore Mode and Execute Mode in Solver Module.

# F    SOLVER MODULE DETAILS

## F.1    MODE SELECTOR

In Section 3.2, we described the Mode Selector when combined with Place Event Memory. When the memory type changes such as Place Memory or Event Memory, different read operation should be employed. We provide the Memory Read operation of Place Memory, and Event Memory in Appendix E.2, and E.3, respectively. In case of FIFO Memory, for read operation, task alignment scores on whole frames in the memory is required.

## F.2    EXPLORE MODE

In explore mode, hierarchical structure is employed. For high-level goal selector, we take the similar approach introduced in Chang et al. (2023), where the target problem is indoor navigation with robots. Their global policy exploits semantic map to output exploration goal. For the goal selection, it uses frontier-based exploration (Yamauchi, 1998), which selects the closest unexplored region as the goal. In Figure 10(a), the overview of hierarchical episodic exploration is illustrated. From Place Event Memory, we contruct a visitation map by marking the agent's visited locations. In addition to marking agent's visited locations, we also marked the agent's FoV in the visitation map with sector region towards agent's head direction (yellow sector shows agent's current FoV). From the visitation map, the next goal is chosen, and the goal, and current observation, position is given to VPT-Nav for generating low-level action. In experiments, for the tasks with map size $100 \times 100$, we used visitation map size $L = 120$, and grid cell size $G = 15$. For the tasks with map size $200 \times 200$, we used visitation map size $L = 240$, and grid cell size $G = 30$.

## F.3    EXECUTE MODE

The model architecture of Execute mode is illustrated in Figure 10(b). When the Mode Selector selects the Execute Mode, the experience frame of a task-relevant resource is retrieved from the memory. MrSteve then navigates to the goal location of the experience frame, and then adjust camera orientation using yaw and pitch from the frame to ensure it faces the observed of the retrieved experience frame again. Once the camera adjustment is complete, Steve-1 is executed to follow the task instruction $\tau_n$.

## F.4    MINECLIP & STEVE-1 PROMPTS FOR TASKS

The prompts used in our experiments are listed in Table 3. MrSteve sets $\tau_n$ to one of two different prompts depending on the agent's status. When the agent is in the mode selection phase, the MineCLIP prompts are used as the query for the memory to determine whether the task-relevant resource exists in the memory. In contrast, the Steve-1 prompts are utilized during the execution mode.

Relying solely on Steve-1 prompts can lead to difficulties in calculating alignment score between the prompt and memory records. Since MineCLIP (Fan et al., 2022b) computes the alignment between videos and textual descriptions, the alignment score is higher when the textual description well reflects the agent's action shown in the video. If a video contains scenes relevant to the current

**Table 3:** Prompts used in MrSteve for each task.

|  | MineCLIP | Steve-1 |
|---|---|---|
| log | near tree | cut a tree |
| beef | near cow | kill cow |
| dirt | near dirt | get dirt |
| sand | near sand | get sand |
| seed | near grass | collect seeds |
| wool | near sheep | kill sheep |
| leaves | near tree | break leaves |
| milk bucket | near cow | get milk bucket |
| water bucket | near water | get water bucket |

task but lacks behavior indicative of task completion, the alignment score will be low. For instance, consider a scenario where the "obtain water bucket 🪣" task is given for the first time, and the agent has previously encountered water. Although the agent is *Near* water in that scene, it has not yet performed the action of *Obtaining* water bucket. As a result, the prompt "obtain water bucket" would not accurately describe that memory, and that scene could be disregarded during the mode selection process. In our experiments, we manually defined two types of prompts for each task. However, this process can be automated using LLMs, which prompts an LLM to extract the more suitable language instructions to complete a task.

## G   EXPLORATION EXPERIMENTS DETAILS

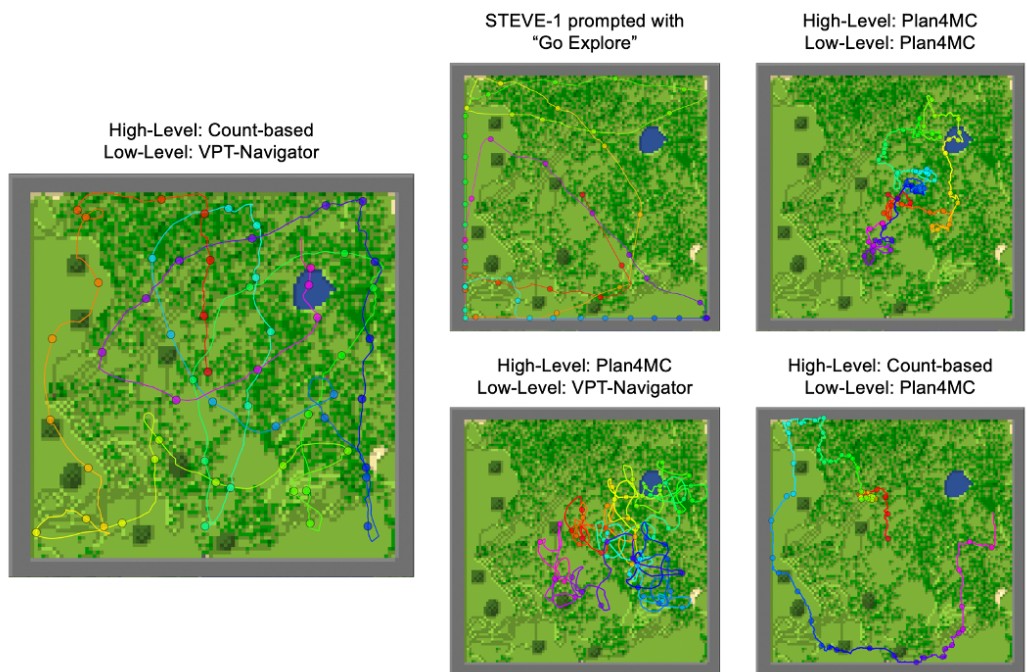

**Figure 11:** Agent's trajectories of length 6K steps on $100 \times 100$ sized map with different exploration methods. Left figure is the agent's trajectory from our exploration method.

We provide further details on evaluation metrics and analysis of the exploration results. For evaluation metrics, we measured Map Coverage and Rivisit Count. Map Coverage is calculated by dividing the map into $11 \times 11$ grid cells and measuring the percentage of cells covered by the agent's trajectory.

Revisit Count measures the average number of times the agent visits the same grid cell only for the cells that have visitation counts larger than $300$. We measured Revisit Count only for highly visited grid cell since the agent needs some amount of time to escape the grid cell.

Since Steve-1 (Lifshitz et al., 2024) has no memory module, it tends to visit the same place multiple times. We also observed that Steve-1 exhibits a behavior of moving straight ahead when task instruction "Go Explore" is given, often colliding with walls and then following along the wall instead of avoiding it, which harms the Map Coverage and Revisit Count.

Plan4MC (Yuan et al., 2023) employs a hierarchical exploration policy. The high-level policy, based on an LSTM model, takes the agent's $(x, y)$ coordinates along its trajectory and outputs the next direction to move (north, south, east, or west). The environment is discretized into an $11 \times 11$ grid, and the policy is trained using PPO (Schulman et al., 2017) to maximize the number of unique grid cells visited. The low-level policy is trained using DQN (Mnih et al., 2015) and follows the MineAgent structure from the MineDojo framework. It receives a goal position and current observation in pixel and outputs actions in the MineDojo action space. However, despite the high-level RNN policy, it struggled to keep track of visited locations as the trajectory grew longer, leading to a high Revisit Count. Furthermore, the low-level DQN controller faced difficulties navigating complex terrain, such as mountains and rivers, resulting in lower Map Coverage.

## H  *ABA-Sparse* TASKS FULL RESULTS

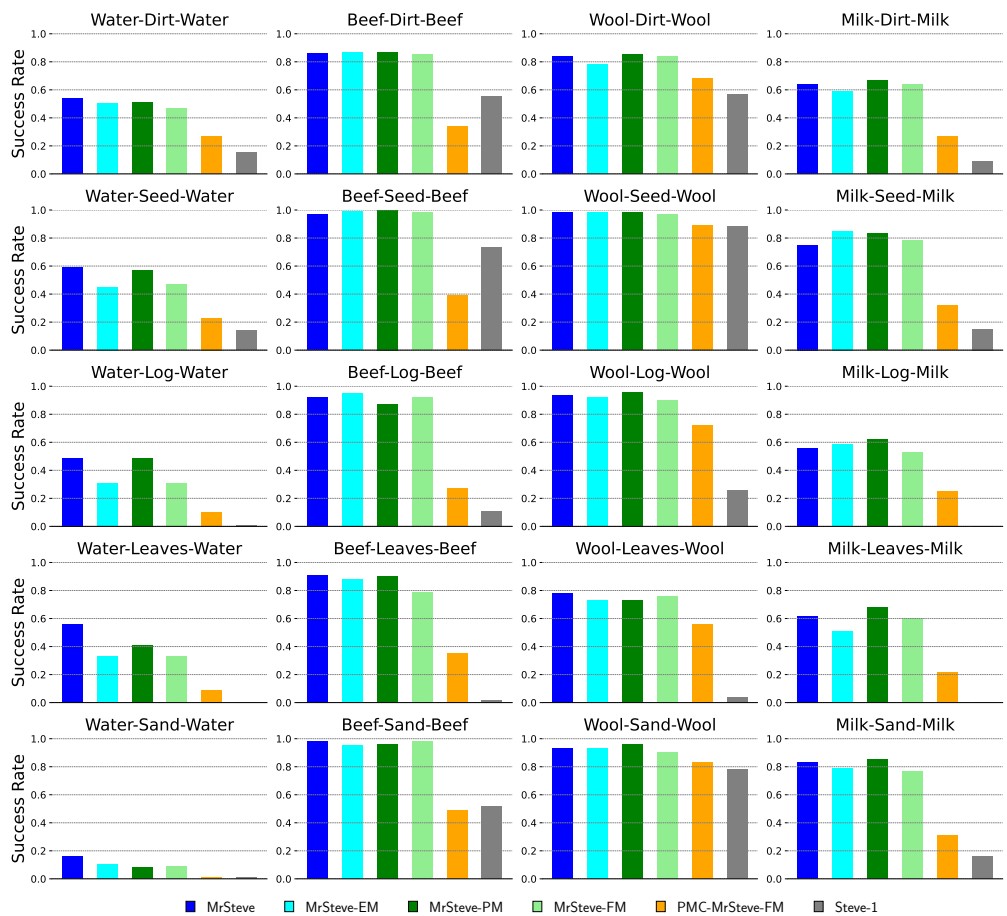

**Figure 12:** *ABA-Sparse* Tasks Full Result.

We report the results of all combinations of the *ABA-Sparse* tasks in Figure 12. The experiments were conducted under the same conditions as described in Section 4.2. We note that MrSteve and

its memory variants consistently outperform Steve-1 in all tasks. In case of PMC-MrSteve-FM, it performed better than Steve-1 in most tasks, while it failed on some tasks due to a suboptimal exploration strategy.

# I    MrSteve in Randomly Generated Maps

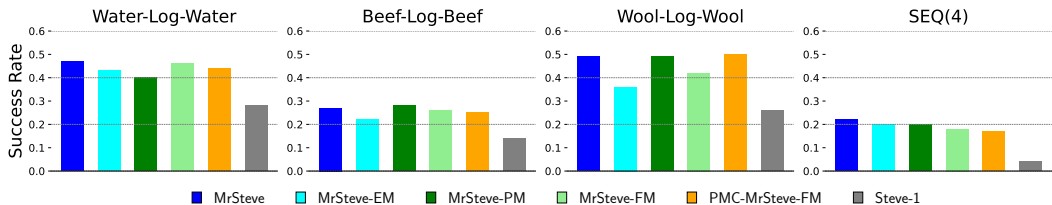

**Figure 13:** The performance of different agents in *ABA-Random* tasks. MrSteve consistently outperforms Steve-1 in randomly generated map.

In this experiment, we investigate whether MrSteve benefits over Steve-1 in sequential task when the map is randomly generated. To verify this, we created random Plains maps using the Minedojo environment (Fan et al., 2022b), and set up tasks similar to the *ABA-Sparse* tasks described in Section 4.2. We call this task, *ABA-Random* task.

In this experiment, task *A* could involve collecting either water, beef, or wool, while task *B* involves gathering logs. Additionally, we introduce a sequential task named `SEQ(4)`, which requires the agent to solve four consecutive tasks: log, water, wool, and beef. For the *ABA-Random* tasks, the agent was given 12K steps, and for the `SEQ(4)` task, 16K steps were allowed. We evaluated the following agents: MrSteve, MrSteve-EM, MrSteve-PM, MrSteve-FM, PMC-MrSteve-FM, and Steve-1.

As shown in Figure 13, MrSteve and its memory variants consistently showed higher success rate than Steve-1 across all tasks. This is because when Steve-1 is finished with task *B*, it tries to solve the final task *A* from scratch, making it difficult to complete all tasks in a limited time. However, MrSteve could retain the experience frames of task *A* in memory, making it efficient to solve all tasks in time. This result suggests that augmenting memory plays a crucial role in solving sequential tasks, even in randomly generated map.

# J    Ablation Studies on Place Event Memory

In this section, we study the robustness of our proposed agent MrSteve by exploring the effects of top-$k$ selection in Mode Selector and the size of a place cluster for Place Event Memory. For this, we used one of the tasks from *ABA-Sparse* task in Section 4.2, which is *Wool-Log-Wool* task. The success rate of MrSteve with different top-$k$'s and place cluster sizes are illustrated in Figure 14. For top-$k$ experiment, we tested five $k$ values, which are 1, 2, 4, 10, and 30. From the results, we found that using small $k$ did not largely affect the performance. For cluster size experiment, we tested five cluster sizes, which are 1, 2, 4, 6, and 12 in Minecraft blocks. From the results, we see that using bigger cluster size slightly lowered the performance since the center embedding of the place cluster may not be good summarization vector of the place when cluster size is large, making the agent difficult to recall task-relevant frames in the memory.

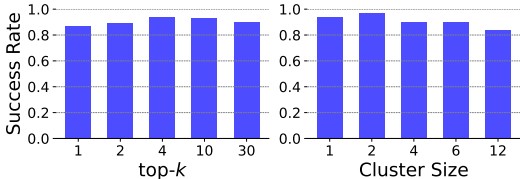

**Figure 14:** Success Rates of MrSteve with different top-$k$'s, and cluster sizes on *Wool-Log-Wool* task.

**Table 4:** Querying Time and FLOPs for each memory type and $k$ values. We report the averages and standard errors over 1K query operations.

| Memory Type | top-$k$ | Time per Query (ms) | FLOPs $(\times 10^9)$ |
|---|---|---|---|
| Place Event Memory | 1 | $6.24 \pm 0.01$ | $1.591 \pm 0.001$ |
| | 2 | $6.47 \pm 0.02$ | $1.610 \pm 0.002$ |
| | 4 | $6.89 \pm 0.02$ | $1.667 \pm 0.003$ |
| | 10 | $7.92 \pm 0.03$ | $1.791 \pm 0.004$ |
| | 30 | $11.24 \pm 0.07$ | $2.210 \pm 0.010$ |
| Event Memory | 1 | $11.28 \pm 0.35$ | $1.603 \pm 0.054$ |
| | 2 | $18.95 \pm 0.56$ | $2.711 \pm 0.073$ |
| | 4 | $42.41 \pm 1.20$ | $3.734 \pm 0.088$ |
| | 10 | $53.82 \pm 1.11$ | $7.476 \pm 0.150$ |
| | 30 | $117.16 \pm 2.02$ | $12.694 \pm 0.201$ |
| Place Memory | 1 | $3.86 \pm 0.05$ | $1.251 \pm 0.001$ |
| | 2 | $4.21 \pm 0.08$ | $1.281 \pm 0.002$ |
| | 4 | $4.72 \pm 0.08$ | $1.338 \pm 0.003$ |
| | 10 | $5.98 \pm 0.09$ | $1.504 \pm 0.005$ |
| | 30 | $9.85 \pm 0.12$ | $2.024 \pm 0.011$ |
| FIFO | – | $438.46 \pm 0.10$ | $52.481 \pm 0.000$ |

## K  COMPUTATIONAL COMPLEXITY OF QUERY OPERATION

We provide the computational complexity of query operation of each memory type in Table 4. To ensure a fair comparison, we generated an episode with a 100K-step trajectory, allowing each type of memory to process identical observations. After constructing each type of memory with this trajectory, we randomly selected 1,000 observation embeddings from the trajectory for the query operations. The query time represents the elapsed time during the query function call. FLOPs denotes the number of floating-point operations required for the MineCLIP score calculation. For FIFO memory, MineCLIP scores are computed once during task alignment score calculation. For the other memory types, MineCLIP scores are calculated twice: during the top-$k$ selection and for the task alignment score.

The number of clusters affects the complexity of top-$k$ selection and the number of frames per cluster influences the complexity of the task alignment score calculation. Place Event Memory consists of approximately 3,000 clusters, with each cluster containing an average of 30 frames. Place Memory, on the other hand, includes about 2,300 clusters, each holding an average of 43 frames. Although Place Memory provides the fastest querying time, Place Event Memory has comparable speed while achieving similar or superior success rates in most tasks.

Event Memory comprises around 582 clusters, with an average of 172 frames per cluster. Its clustering approach relies on visual discriminative features, which means visually similar places are grouped together, even if they are spatially distant. As a result, Event Memory has significantly fewer clusters compared to Place and Place Event Memory.

Overall, the three hierarchical memory types are computationally efficient and lightweight compared to FIFO Memory, which calculates the task alignment scores on whole 100K frames. In contrast, Place Event Memory with $k = 30$ evaluates roughly 3,900 frames per query (3,000 for top-$k$ selection and 30 frames per cluster across 30 clusters), resulting in a performance that is approximately 40 times faster and requires about 24 times fewer FLOPs than FIFO Memory.

## L  GOAL-CONDITIONED VPT NAVIGATOR DETAILS AND INVESTIGATION

### L.1  GOAL-CONDITIONED VPT NAVIGATOR FINE-TUNING DETAILS

When the goal location $l_G$ is selected by high-level goal selector, it is important for the agent to navigate to the goal location accurately. Since navigating complex terrains (*e.g.*, river, mountain) requires human prior knowledge, we use VPT as our initial policy, and fine-tune it for goal-conditioned

navigation policy. We name this model as VPT-Nav. To see how VPT-Nav model works, we first describe the components of VPT as follows:

$$
\begin{aligned}
\text{Image Encoder:} \quad & x_t = I_\theta^{\mathcal{E}}(i_t) \\
\text{Transformer-XL:} \quad & z_{t-T:t} = \text{TrXL}_\theta(x_{t-T}, \cdots, x_t) \\
\text{Policy Head:} \quad & \hat{a}_t \sim \pi_\theta(a_t | z_t)
\end{aligned}
\tag{1}
$$

In previous works, there were different approaches in fine-tuning VPT for specific purpose. First is goal-conditioned behavior cloning. In Steve-1 (Lifshitz et al., 2024), authors added linear projection of a text embedding to the image embedding $x_t$ and fine-tuned the whole VPT model for behavior cloning from human demonstration data. This makes Steve-1 a text-conditioned VPT. In GROOT (Cai et al., 2023b), authors used gated cross-attention layers (Alayrac et al., 2022) in Transformer-XL (Tr-XL) to condition the video of some tasks (*e.g.*, log wood). GROOT was trained for behavior cloning from reference videos of human plays working as video-conditioned VPT.

The second is RL fine-tuning for the single task. In DECKARD (Nottingham et al., 2023) and PTGM (Yuan et al., 2024), VPT was fine-tuned for single task with PPO (Schulman et al., 2017) by attaching adaptor (Houlsby et al., 2019) in Tr-XL layers, and value head $\hat{v}_t = v_\theta(z_t)$. When fine-tuning, only the adaptors, policy head, and value head were updated. For learning stability, those works used different KL loss in PPO objective, which is KL loss between fine-tuning policy and VPT policy to keep the VPT's prior knowledge.

Since our target is to train goal-conditioned navigation policy, one way to achieve this is to naively combining ideas from 1) goal-conditioned behavior cloning, and 2) RL fine-tuning for the single task. We first make goal embedding $G(l_t, l_G)$ from the agent's location $l_t$ and goal location $l_G$ with goal encoder $G(\cdot)$, then add it to image embedding $x_t$. Second, we attach the adaptor in Tr-XL layers, and value head. Then, we fine-tune the whole model or only adaptors and policy, value heads with PPO.

However, we found that this naive approach showed suboptimal navigation behavior in complex terrains such as mountain and river. We speculated that this is because RL objective is rather weak learning signal compared to behavior cloning, which causes hardship in giving information of goal embedding to the policy head. Also, some information of goal embedding may be corrupted while it is added to image embedding, and processed by Tr-XL layers. Thus, we made the following modifications in the model architecture. First, instead of giving goal embedding in Tr-XL input, we added the goal embedding to policy head input. Second, we used recently proposed adaptor for Tr-XL, which is LoRA (Hu et al., 2021b). The following changes enabled VPT-Nav to exhibit optimal navigation behavior. We provide an investigation of this model architecture search in Appendix L.2.

With the changes in previous paragraph, the modified VPT for navigation with the learning parameters $\psi$ has the following components:

$$
\begin{aligned}
\text{Image Encoder:} \quad & x_t = I_\theta^{\mathcal{E}}(i_t) \\
\text{Transformer-XL:} \quad & z_{t-T:t} = \text{TrXL}_\psi(x_{t-T}, \cdots, x_t) \\
\text{Goal Conditioning:} \quad & z_t' = G_\psi^{\mathcal{E}}(l_t, l_G) + z_t \\
\text{Policy Head:} \quad & \hat{a}_t \sim \pi_\psi(a_t | z_t') \\
\text{Value Head:} \quad & \hat{v}_t = v_\psi(z_t'),
\end{aligned}
\tag{2}
$$

Here, goal encoder $G_\psi^{\mathcal{E}}$ is 4-layer MLP, and value head is a randomly initialized single linear layer. Parameter $\psi$ in Tr-XL indicates LoRA parameters. We use PPO for training with reward based on the increase or decrease in Euclidean distance between locations of the goal and the agent. When the agent reaches the goal location within 3 blocks, it is considered a success, and an additional reward of 100 is given. The detailed hyper-parameters for training will be found in Table 5.

## L.2 MODEL ARCHITECTURE SEARCH FOR GOAL-CONDITIONED VPT NAVIGATOR

In the previous section, we discussed the naive way of combining the idea of goal-conditioned behavior cloning and RL finetuning for training goal-conditioned navigator. In this section, we conduct a model architecture search on VPT model to find the optimal goal-conditioned navigation model. To do this, we focused on three key design choices: 1) the input location of goal embedding, 2) training parameters, and 3) different Tr-XL adaptors. For the location of goal embedding, we

**Table 5:** Hyper-parameters for the Goal-Conditioned Navigation VPT Training.

| Name | Value |
| --- | --- |
| Initial VPT Model | `rl-from-foundation-2x` |
| Discount Factor | 0.999 |
| Rollout Buffer Size | 40 |
| Training Epochs per Iteration | 5 |
| Vectorized Environments | 4 |
| Learning Rate | $10^{-4}$ |
| KL Loss Coefficient | $10^{-4}$ |
| KL Loss Coefficient Decay | 0.999 |
| Total Iteration | 400K |
| Steps per Iteration | 500 |
| GAE Lambda | 0.95 |
| Clip Range | 0.2 |

consider the three tactics: (1) add the goal embedding to the image embeddings and input to Tr-XL (**TrXL-Cond.**), (2) add the goal embedding to Tr-XL output (**Head-Cond.**), and (3) add the goal embedding both at the input and output of Tr-XL (**Dual-Cond.**). For training parameters, we use full-finetuning or finetuning only part of the model. For the Tr-XL adaptors, we consider the adaptor from Houlsby et al. (2019), and LoRA adaptor (Hu et al., 2021b). In Figure 15, we illustrate the different model architectures for goal-conditioned VPT finetuning models.

For the model search, we summarized the navigation performance (SPL and Success Rate (SR)) of different model architectures in Table 6. Interestingly, we found that adding goal embedding to image embeddings (**TrXL-Cond.**) showed suboptimal behavior, while adding goal embedding to output of Tr-XL (**Head-Cond.**, **Dual-Cond.**) showed good performance indicating that propagating learning signal to goal embedding through Tr-XL is difficult from RL objective. We note that using LoRA adaptor showed higher performance than adaptors from Houlsby et al. (2019). In conclusion, using **Head-Cond.** with LoRA finetuning performed the best. Additionally, we tested one additional model that does not update Tr-XL, which showed comparable result to the best performing model. This architecture benefits from using a smaller number of learning parameters and lower gradient computations. Thus, we use the VPT-Nav trained with one of two architectures, which are **Head-Cond.** with LoRA finetuning and **Head-Cond.** with No Tr-XL Update.

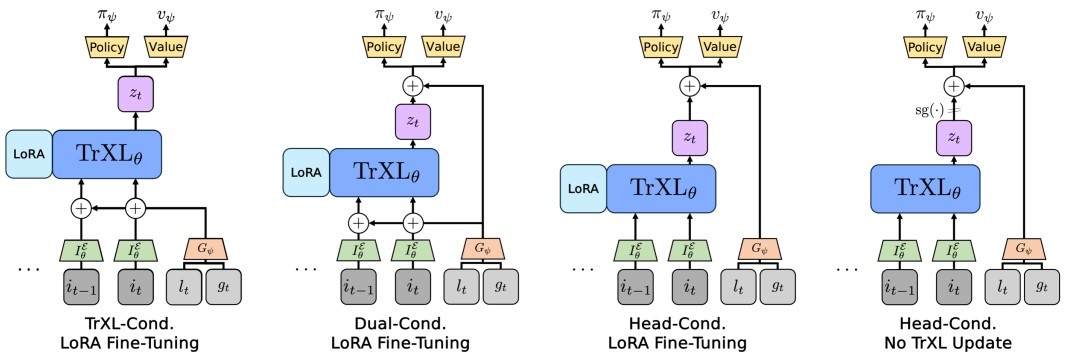

**Figure 15:** Four Types of Navigation Goal-Conditioning.

### L.3 VPT NAVIGATOR ABLATION STUDY

We investigate how KL loss in PPO objective (*i.e.*, KL loss between fine-tuning VPT and original VPT) affects the performance of VPT-Nav in diverse environments (Table 7). We measured the SPL and Success Rate (SR) of the navigators in navigation tasks where the goal location is within 10–20 blocks away from agent's location. While Heuristic method performs best in Flat, Plains, and Mountain tasks, VPT-Nav with KL coefficient $10^{-4}$ showed high performances in all tasks. In the case of a low-level navigator in Plan4MC (Yuan et al., 2023), which uses DQN policy, we observed

**Table 6:** VPT-Nav Performance of different Goal-Conditioning Methods. Top-1 performances are bolded.

| Conditioning | Fine-tuning | Flat | Plains | Mountain | River |
|---|---|---|---|---|---|
| TrXL | Full | 0.050 (5%) | 0.423 (46%) | 0.000 (0%) | 0.000 (0%) |
| | Houlsby et al. (2019) | 0.488 (56%) | 0.305 (34%) | 0.052 (32%) | 0.000 (0%) |
| | LoRA | 0.481 (55%) | 0.475 (51%) | 0.058 (42%) | 0.000 (0%) |
| Dual | Houlsby et al. (2019) | 0.883 (95%) | 0.828 (90%) | 0.125 (94%) | 0.066 (31%) |
| | LoRA | 0.798 (96%) | 0.762 (90%) | **0.169 (100%)** | 0.287 (**100%**) |
| Head | No TrXL Update | 0.849 (96%) | 0.780 (88%) | 0.157 (**100%**) | 0.274 (87%) |
| | Houlsby et al. (2019) | 0.729 (85%) | 0.841 (91%) | 0.159 (**100%**) | 0.052 (23%) |
| | LoRA | **0.904 (98%)** | **0.880 (95%)** | 0.150 (**100%**) | **0.307 (100%)** |

suboptimal behavior. We note that VPT-Nav is robust to tasks with difficult terrains such as Mountain and River since VPT (Baker et al., 2022) is trained from human demonstration data, which has high-quality navigation ability. Also, using a large KL coefficient (*e.g.*, $10^{-2}$) harmed the overall performance, while using a small KL coefficient (*e.g.*, 0) harmed the robustness of the navigator in complex tasks (*e.g.*, River).

**Table 7:** Performance of different Low-Level Navigators. Top-2 performances are bolded.

| SPL (Success Rate) | Flat | Plains | Mountain | River |
|---|---|---|---|---|
| Heuristic | **0.962 (100%)** | **0.906 (94%)** | **0.188 (100%)** | 0.000 (0%) |
| Low-level Navigator in Plan4MC | 0.416 (61%) | 0.326 (47%) | 0.010 (10%) | 0.000 (0%) |
| VPT-Nav (KL = 0) | 0.806 (90%) | 0.774 (85%) | 0.131 (97%) | **0.034 (13%)** |
| VPT-Nav (KL = $10^{-4}$) | **0.849 (96%)** | **0.780 (88%)** | **0.157 (100%)** | **0.274 (87%)** |
| VPT-Nav (KL = $10^{-3}$) | 0.762 (95%) | 0.651 (78%) | 0.015 (17%) | 0.012 (6%) |
| VPT-Nav (KL = $10^{-2}$) | 0.692 (84%) | 0.702 (83%) | 0.078 (87%) | 0.014 (7%) |

# M    LLM-AUGMENTED AGENT WITH MRSTEVE

**Table 8:** Long-Horizon Planning Tasks details. The required subgoals denote the length of the shortest plan for each task. The items that the low-level controller should obtain are listed in Items to Obtain column.

| Task | Max Steps | Required Subgoals | Items to Obtain |
|------|-----------|-------------------|------------------|
| oak_stairs | 3000 | 4 | log×3 |
| sign | 3000 | 5 | log×3 |
| fence | 3000 | 5 | log×3 |
| bed | 6000 | 5 | log×3, wool×3 |
| painting | 6000 | 6 | log×2, wool×1 |
| carpet | 6000 | 2 | wool×2 |
| item_frame | 6000 | 6 | log×2, leather×1 |
| leather_boots | 6000 | 5 | log×1, leather×4 |
| leather_chestplate | 6000 | 5 | log×1, leather×8 |
| leather_helmet | 6000 | 5 | log×1, leather×5 |
| leather_leggings | 6000 | 5 | log×1, leather×7 |

**Table 9:** Success Rate of two low-level controllers with DEPS (Wang et al., 2023c) planner.

| Task | Steve-1 | MrSteve |
|------|---------|---------|
| oak_stairs | 67% | 80% |
| sign | 53% | 60% |
| fence | 40% | 50% |
| bed | 27% | 50% |

In this section, we investigate synergy between MrSteve and LLM-augmented hierarchical agent framework with long-horizon planning tasks listed in Table 8. We follow DEPS (Wang et al., 2023c) as LLM-augmented high-level planner with two low-level controllers: Steve-1 and MrSteve.

Once the target object for a task is given, the high-level planner asks the LLM to make the initial plan $\mathcal{P}_t = \{\tau_{t,i}^g\}_{i=1}^N$ for the task, where $\tau_{t,i}^g$ is $i$-th subgoal at timestep $t$ in the textual form. After the initial plan is given, the low-level controller executes the subgoal sequentially. Each subgoal is `mine` or `craft` type, where `craft` subgoals are executed heuristically via MineDojo functional actions, whereas `mine` subgoals are executed by low-level controllers.

LLM-generated initial plans can be inaccurate, so the low-level controllers often fail to execute subgoals. For instance, we found that the initial plans frequently omit the subgoal for creating crafting table, which is prerequisite item for the tasks in Table 8 except the carpet task. To address this problem, DEPS framework introduced the replanning procedure with the descriptor and explainer modules. The descriptor module makes a text prompt representing the inventory of the agent. The explainer modules ask the LLM the reason of failure based on the text prompt from the descriptor module. Based on the explanation, the high-level planner asks the LLM to revise the plan and the low-level controller executes subgoals from the revised plan.

Additionally, because the LLM does not observe the environment, the order of subgoals in a plan can be suboptimal. When some subgoals share the same prerequisite so they can be executed in any order, it is more efficient to execute subgoal, which can be completed faster than the other subgoals, first. This can prevent wasting time by giving up subgoals that could be completed quickly, pursuing a subgoal whose completion time is uncertain, and then going back to search for the previously quicker task again after executing the subgoal. In DEPS, this procedure called elector module is implemented with the horizon prediction module in GSB (Cai et al., 2023a). The horizon prediction module $\mu(s_t, \tau_{t,i}^g)$ takes the current observation $s_t$ and the textual subgoals $\{\tau_{t,i}^g\}_{i=1}^N$ and predicts the time to complete each subgoal. Based on the time prediction, a subgoal to be executed is sampled from the distribution as follows:

$$\text{Selector}(\tau_{t,i}^g; s_t, \mathcal{P}_t) = \frac{\exp(-\mu(s_t, \tau_{t,i}^g))}{\sum_j \exp(-\mu(s_t, \tau_{t,j}^g))}. \tag{3}$$

**Table 10:** Success Rate of two low-level controllers with the Ground-Truth plan for each task.

| Task | Steve-1 | MrSteve |
|---|---|---|
| oak_stairs | 80% | 83% |
| sign | 70% | 67% |
| fence | 67% | 70% |
| bed | 37% | 60% |
| painting | 60% | 73% |
| carpet | 43% | 60% |
| item_frame | 53% | 63% |
| leather_boots | 13% | 33% |
| leather_chestplate | 3% | 17% |
| leather_helmet | 20% | 20% |
| leather_leggings | 0% | 13% |

We report success rates of DEPS framework with Steve-1 and MrSteve in Table 9. We use Qwen2.5-72B (Team, 2024) as LLM in the high-level planner. DEPS with MrSteve shows comparable performance or outperforms compared to DEPS with Steve-1. Especially, there is a huge performance gap between the two low-level controllers in the bed task, which requires killing three sheeps. We observe that when the agent hit a sheep, it runs away from the agent and the agent chases it until the agent gets wool items. After this, Steve-1 easily forgets the place where other sheeps exist and tries to find sheep. However, MrSteve avoids this redundant exploration by utilizing the episodic memory. Hence, these results highlight the importance of episodic memory for low-level controllers in LLM-augmented hierarchical agent frameworks.

In Table 10, we also report success rates of hierarchical agents with Steve-1 and MrSteve and the optimal plan for each plan. In this setting, because there is no instability from the LLM, the performance is solely determined by performance of low-level controllers. Based on this experiment, although the goal is optimal, Steve-1 shows lower performance compared to MrSteve, indicating low-level controllers is a performance bottleneck of hierarchical agent frameworks.

## N  TASK-CONDITIONED HIERARCHICAL EPISODIC EXPLORATION

**Table 11:** Performance Comparison of Two Exploration Methods. We report success rates and the average and standard error of execution times for the explore mode in *ABA-Sparse* tasks.

|  | Task-Conditioned Exploration | | Task-Free Exploration | |
|---|---|---|---|---|
|  | Success Rate | Explore Mode Length | Success Rate | Explore Mode Length |
| Beef-Log-Beef | 93% | $1202.64 \pm 67.49$ | 92% | $1512.00 \pm 62.37$ |
| Beef-Leaves-Beef | 90% | $1269.23 \pm 66.57$ | 91% | $1496.10 \pm 69.43$ |
| Wool-Sand-Wool | 98% | $924.98 \pm 56.85$ | 93% | $1350.00 \pm 78.80$ |
| Wool-Dirt-Wool | 84% | $1303.21 \pm 117.04$ | 84% | $1452.00 \pm 89.92$ |
| Milk-Sand-Milk | 86% | $1197.29 \pm 71.44$ | 83% | $1526.48 \pm 63.31$ |
| Milk-Leaves-Milk | 59% | $1224.24 \pm 68.79$ | 62% | $1579.35 \pm 64.61$ |

In this section, we investigate an advanced version of hierarchical episodic exploration. The current exploration method selects the next exploring position based on the visitation map generated from PEM, while it does not use the information inherited in PEM. Hence, the current exploration method is task-free exploration. While not using information from PEM worked pretty well in practice, using knowledge from PEM may have better explorative behaviors. Thus, we implemented a new task-conditioned exploration method which exploits information stored in PEM and evaluated two exploration methods on exploration and task-solving tasks.

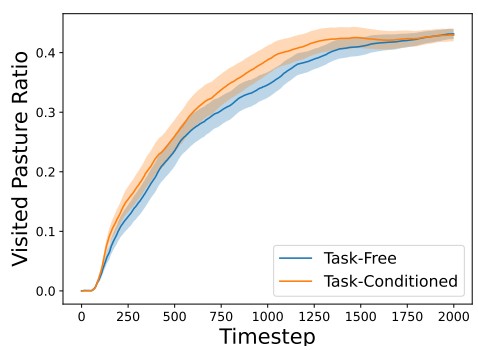

**Figure 16:** Given the task is "find cow", the ratio of pastures among the locations explored over time is presented for two agents, each employing a different exploration method.

The two exploration methods have the same primary goal: to visit the least visited places first. The difference between the two exploration methods is how to select one of the least-visited places. The task-free exploration method randomly selects the next exploring position among the least-visited locations. In contrast, the task-conditioned exploration method selects the next exploring position among the least-visited locations that are estimated to be related to the task based on the information inherited in PEM.

To implement the task-conditioned exploration method, the high-level exploration policy first accesses all event clusters when selecting the next exploring position. Using MineCLIP, the task-relevant score for each event cluster is estimated by calculating the alignment score between the center embedding of the event cluster and the text prompt. If we use the text prompts for MineCLIP listed in Table 3, the alignment score for locations where the target object has not yet been observed will tend to be low, even if the location is relevant to the task, potentially hindering the exploration of those areas.

To address this problem, we used text prompts that describe places related to the task. For instance, the text prompt for log 🟫 is set to "near pasture", while the text prompt for sand 🟫 is set to "near desert." After that, Event clusters with alignment scores exceeding the threshold are collected, and a task-relevance map is constructed to represent the agent's FoVs of these event clusters, similarly to the visitation map building method described in Appendix F.2.

Additionally, the $3 \times 3$ box blur filter is applied to the task-relevance map to introduce an inductive bias, assuming that if something is near $X$, it is likely to also be $X$. Finally, based on the task-relevance map, the high-level exploration policy selects the next exploring position by prioritizing locations that are the least visited, most task-relevant, and closest to the agent.

Figure 16 shows the proportion of task-relevant locations among the places explored by the agent, demonstrating that the task-conditioned exploration method explores task-relevant locations more quickly in the early stages. We used Map 2 in Figure 8 and set the task as "find cow," and therefore measured the proportion of pastures, which are relevant to this task. In the early stages of the episode, the task-conditioned exploration method explored pastures more than the task-free exploration method. After approximately 2000 steps, however, the exploration tendencies of both methods became similar.

Table 11 shows performance comparison between the two exploration methods in *ABA-Sparse* tasks. We report success rates and execution times for the explore mode from MrSteve with the task-free and task-conditioned exploration methods. The success rates between the two exploration methods are comparable, while the task-conditioned exploration is finished earlier than the task-free exploration.

Through the results of the two experiments, we confirmed that fully utilizing PEM not only aids in task-solving but also optimizes exploration more effectively.

## O  *ABA-Sparse* TASKS WITH MEMORY CONSTRAINTS

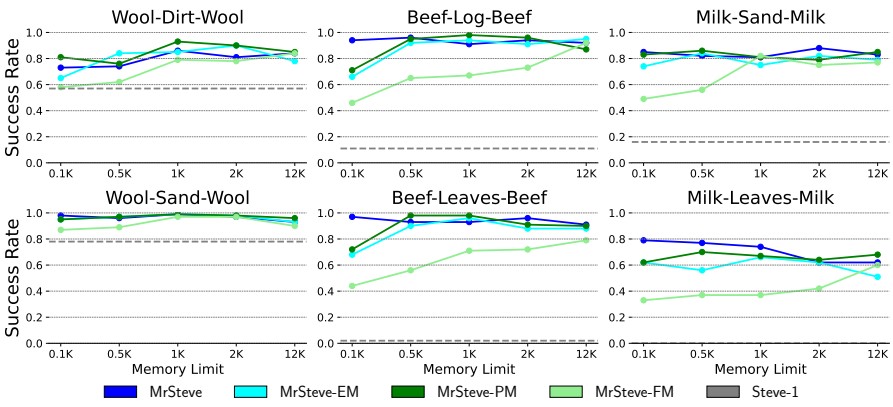

**Figure 17:** Success Rate Comparison between MrSteve with its memory variants and Steve-1 in *ABA-Sparse* tasks with memory constraints. We tested different memory capacities (0.1K to 12K) for each model. In all tasks, the performance of MrSteve-FM decreases as memory capacity gets smaller. We note that MrSteve is robust to memory capacities across tasks, while MrSteve-PM, and MrSteve-EM showed performance degradation in Beef-Log-Beef and Beef-Leaves-Beef tasks.

In this section, we investigate how MrSteve and its memory variants perform in *ABA-Sparse* tasks in Section 4.2 when memory capacity is limited. We evaluated each model with different memory capacities ranging from 0.1K to 12K, which is the maximum episode length. In all tasks, we observe that the performance of MrSteve-FM decreases as the memory capacity gets small. This is because FIFO Memory in MrSteve-FM losts relevant frames in first task *A* while solving task *B*. While MrSteve showed robust performance to constrained memory capacities across tasks, MrSteve-PM, and MrSteve-EM showed degraded performances in Beef-Log-Beef and Beef-Leaves-Beef tasks when memory capacity is 0.1K. This indicates the robustness of PEM to memory capacities in *ABA-Sparse* tasks.

## P  PLACE EVENT MEMORY INVESTIGATION

Although our place event memory enables efficient querying by clustering experience frames, it stores not only the center embedding of each event cluster but also all the frames that constitute the event clusters. Since the frames within each cluster contain highly similar information, storing all of them can increase redundancy in the memory system, potentially degrading storage efficiency and retrieval performance. Therefore, we attempted to optimize the memory-storing method of PEM in the simplest way possible to investigate whether reducing the storage of redundant information could achieve better efficiency.

We implemented the modified PEM, which stores only center embeddings. The write and read operations of the modified PEM are the same as the original PEM. However, once event clustering is executed, the modified PEM stores only the center embedding of each event cluster and removes the embeddings of other frames. In our experiment settings, event clustering is performed when the dummy deque of a place cluster has 100 frames, and until then, frames are stored in the dummy deque of a place cluster, with each place cluster capable of holding up to 100 frames. However, even

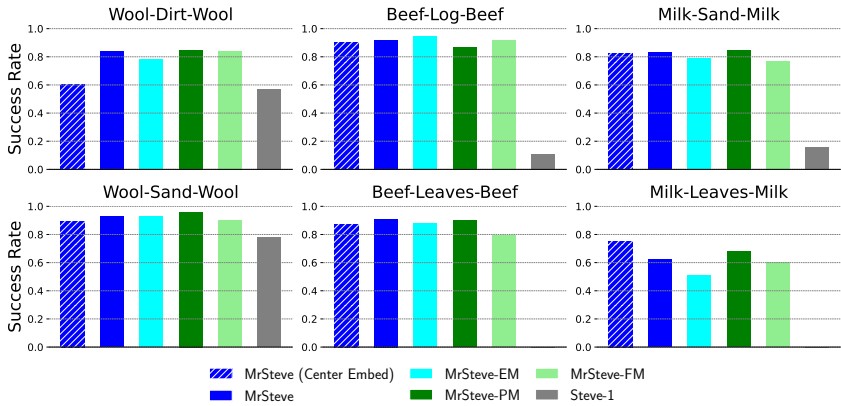

**Figure 18:** Success Rates Comparison between MrSteve with its memory variants and Steve-1 in *ABA-Sparse* tasks. We additionally evaluated MrSteve with modified place event memory, named MrSteve (Center Embed), that stores center embeddings only.

in such cases, memory read operation is highly optimized by accessing only the oldest frame in each place cluster during querying operations.

Figure 18 shows the performance comparison between MrSteve with its memory variants and Steve-1 in *ABA-Sparse* tasks. Surprisingly, the performance of the simplest optimized PEM, named MrSteve (Center Embed), was not dropped significantly. Nevertheless, the computational cost for the querying operation can be reduced. After 12K environmental steps, $305.52$ event clusters, on average, were generated across all tasks, with a standard error of $4.98$. The original PEM stores all 12K frames and each event cluster in PEM holds approximately 40 frames on average. In the end, PEM calculates the alignment scores between frames from the top-30 relevant event clusters and the task instruction, resulting in 1.2K comparisons. However, the modified PEM calculates the alignment scores for around 0.3K frames, making it more efficient.

This experiment result demonstrates the potential to further optimize the PEM memory-storing method. However, our simplest approach resulted in a slight performance drop. Hence, optimizing memory to further reduce redundancy without losing important frames is a worthwhile direction for future work.

