# OpenReview forum: "MrSteve: Instruction-Following Agents in Minecraft with What-Where-When Memory"
_ICLR.cc/2025/Conference — ICLR 2025 Poster_

### Official Review · Reviewer_86n7 · 2024-10-23

**Soundness:** 3
**Presentation:** 3
**Contribution:** 2
**Rating:** 6
**Confidence:** 4

**Summary:**

This work approaches the failures of low-level controllers for navigation in embodied AI environments. It argues that one of the crucial reasons for these failures is the lack of memory for recalling objects and events, which could lead to redundant exploration. For this purpose, it proposes Place Event Memory (PEM), an episodic memory mechanism that tracks entities alongside with their locations and associated events. By using PEM, they employ a count-based exploration strategy, which alternates exploration/exploitation behaviors based on the recalled events. The work also proposes a new low-level navigator built on top of the TrXL architecture. Experiments in simple minecrafts navigation tasks demonstrate gains over prior methods and baselines with either no or simpler mechanisms
of memory.

**Strengths:**

- The work is well written and structured, easy to follow, with good illustrations; the proposed method and motivation are clear.

- The particular choice of tasks is very smart since it shows different scenarios where only place memory or only event memory are not enough, justifying a combined mechanism as proposed by the work. This provides good positive evidence for the scenarios where the method is effective.

- The evaluation setup also comprises different setups and evaluation criteria to evaluate the method in terms of exploration and under circumstances of sparsity, constrained memory, and long-horizon navigation.

**Weaknesses:**

- The major criticism comes from the fact that the paper is motivated by advancing methods for general-purpose embodied AI, but the evaluated scenarios are simplified versions of the SLAM [1] problem. The only considered task is navigation, and the work assumes access to a grid map with the perfect localization of entities. Entities are also assumed to be static (see the point below). In this setup, the only challenge seems to be identifying the different objects, which is actually easily solvable by visual foundational models (MineCLIP). In this setup, one could easily build and maintain a topological map of the environment as in prior work [2, 3] and perform exploration and navigation on top of it. Therefore, it is unclear whether (1) the tasks are challenging enough to reflect general-purpose embodied AI problems and (2) what are the advantages of the proposed method over classic SLAM methods. These points should be clarified in the paper.

- As mentioned in the previous point, entities are assumed to be static, which assumes the entities in the map have fixed location and state. Nevertheless, in open-world settings this does not hold true, i.e., a “zombie” may move around or “be burnt” while the agent is not observing. These changes in the entities would add an uncertainty component in the memory that is disregarded by the proposed method - in other words, memory may become stale over time and the proposed mechanism does not have a way to update/discard these memories. Again, the work should clarify this potential limitation.

- The Success Rate bar plots (in Figures 5, 6, 12, 13, 14) should bring the error bars (confidence intervals) so that it is possible to analyze statistical significance of the reported results.

**Questions:**

As mentioned in my points above:

- What are the advantages of the proposed method in comparison with other methods for the SLAM/robot navigation problem?

- How does the proposed method function in the case of non-static entities, as pointed out in my second concern?


**Summary of the Review**:

The work is well written, the method is very clearly presented, and the evaluation setup is well diversified, covering important aspects of what is proposed. Nonetheless, it is unclear if the considered navigation tasks are challenging enough to reflect the adopted motivation. It is also unclear if the method is advantageous over the classic robot navigation method and how the work is placed in comparison with this literature. I believe these are crucial questions to be answered by the work in order to understand what we can take from it.

**References**

[1] Simultaneous localization and mapping (SLAM). Wikipedia, available in: https://en.wikipedia.org/wiki/Simultaneous_localization_and_mapping

[2] Street et. al. Multi-Robot Planning Under Uncertainty with Congestion-Aware Models. AAMAS, 2020.

[3] Garg et. al. RoboHop: Segment-based Topological Map Representation for Open-World Visual Navigation. ICRA, 2024.

**POST-REBUTTAL**

I appreciate the authors' efforts on the rebuttal phase to clarify reviewers' concerns.
After reading the comments, I believe my concerns were only partially addressed, particularly the first concern related to the comparison with other approaches for SLAM. The rebuttal limited to contrast with other recent work, but without any empirical comparison nor considering classic work on the Robotics literature.  My second concern was acknowledged by the authors during the rebuttal, and I understand it is a natural limitation of the work to be addressed in future iterations. My third concern was not addressed and was unclear why it was not possible to use the 100 seeds already executed to compute the error bars.

I am slightly increasing my score to borderline acceptance to reflect the additional evidence and clarifications raised by the authors in response to all reviewers. Still, I recommend authors to consider classic SLAM approaches as future baselines for this line of work. This weakness is a consensus among the reviewers.

---

> ### Author Response · Authors · 2024-11-25
>
> We sincerely appreciate the reviewer for positive feedback about our paper and experiment setup and valuable comments for our paper’s weakness.
>
> ## Aren’t the tasks, considering only navigation-like tasks, too simple?
>
> We sincerely thank the reviewer for thoughtful feedback on task complexity. We would like to emphasize that Minecraft has long been regarded as a benchmark environment for general-purpose embodied AI due to its inherently complex and dynamic environment. In our experiments, the *ABA-Sparse* tasks in Section 4.2 and the Long-Instruction task in Section 4.4 involve interactions with dynamic entities such as cows and sheep, demonstrating that these tasks go beyond mere navigation challenges. Additionally, we would like to highlight our core contribution: the demonstrated failure of Steve-1 in instruction-following scenarios, as shown in Figure 1 of the manuscript, and further elaborated in Common Response **CR1**.
>
> For further clarity, we have provided two demonstration videos: one task showcasing an *ABA-Sparse* task (milk-leaves-milk) and another illustrating long-horizon planning task using LLM with Mr. Steve (make a bed). These videos are available at https://imgur.com/a/brKrQhL.
>
> ## What are the advantages of the proposed method over classic SLAM?
>
> We thank the reviewer for their constructive comments regarding the relation between with our exploration method and methods from robotics domain. We answered this in Common Response **CR3**.
>
> ## Limitations due to the lack of memory update/discard mechanism
>
> Thanks for the constructive feedback. We acknowledge that PEM has limitations on storing dynamic entities. Although Mr.Steve shows promising performance in tasks requiring interaction with those dynamic entities (*e.g.*, *ABA-Sparse* Milk-Leaves-Milk task), the memory in current PEM can become stale over time, posing difficulties in discarding or updating the memory. We discussed this issue in Common Response **CR0**.
>
> ## Error Bars in Success Rate Plots
>
> We appreciate the valuable feedback regarding statistical significance analysis. We acknowledge that including error bars is important for analyzing statistical significance. In our experiments, we followed the evaluation procedures of previous works [1, 2], which evaluate tasks in Minecraft using 30 to 70 seeds per task. To ensure robustness, we went beyond this standard by running all tasks with 100 different seeds to compute the success rates. However, due to the slow simulation speed of Minecraft and computational constraints, it was not feasible to perform additional runs beyond this setup during the discussion period. Nevertheless, we understand the importance of this aspect and will include results with more seeds and corresponding confidence intervals in the camera-ready version of the paper.
>
> **References**
>
> [1] Zihao Wang, *et al*. “Describe, explain, plan and select: Interactive planning with large language models enables open-world multi-task agents.” NeurIPS. 2023.
>
> [2] Zihao Wang, *et al*. “JARVIS-1: Open-world Multi-task Agents with Memory-Augmented Multimodal Language Models.” NeurIPS Workshop on Agent Learning in Open-Endedness. 2023.

---

> > ### Comment · Reviewer_86n7 · 2024-11-28
> > **Thank you for your rebuttal**
> >
> > Dear authors,
> >
> > I appreciate the authors' efforts on the rebuttal phase to clarify reviewers' concerns. After reading the comments, I believe my concerns were only partially addressed, particularly the first concern related to the comparison with other approaches for SLAM. The rebuttal limited to contrast with other recent work, but without any empirical comparison nor considering classic work on the Robotics literature. My second concern was acknowledged by the authors during the rebuttal, and I understand it is a natural limitation of the work to be addressed in future iterations. My third concern was not addressed and was unclear why it was not possible to use the 100 seeds already executed to compute the error bars.
> >
> > I am slightly increasing my score to borderline acceptance to reflect the additional evidence and clarifications raised by the authors in response to all reviewers. Still, I recommend authors to consider classic SLAM approaches as future baselines for this line of work. This weakness is a consensus among the reviewers.

---

> > > ### Author Response · Authors · 2024-12-04
> > >
> > > We sincerely thank the reviewer for the constructive feedback and for patiently waiting for our response. We hope that our clarifications address your concerns effectively.
> > >
> > > ## Comparison with SLAM-based approaches
> > > As suggested, we have conducted a comparison study between the PEM and a SLAM-based approach from the robotics literature. The results of these experiments are detailed in Common Response **CR4**.
> > >
> > > ## It is unclear why it was not possible to compute error bars of Success Rates
> > > Following your suggestion, we computed error bars for all figures in the manuscript by dividing the 100 seeds into 5 chunks and calculating the standard deviation of the average performance across these chunks. The updated figures are provided at https://imgur.com/a/e2bkUot, and we will update those figures in the camera-ready version of the paper.

---

### Official Review · Reviewer_UxTw · 2024-10-28

**Soundness:** 3
**Presentation:** 3
**Contribution:** 2
**Rating:** 6
**Confidence:** 4

**Summary:**

The paper tries to address the forgetting problem in long-horizon embodied control tasks, such as Minecraft, by introducing a new episodic-memory approach called ‘Place Event Memory’ (PEM). This method organizes past observations based on their locations within the environment (Place Memory). To minimize memory redundancy, each group within the Place Memory is further refined into an event-based memory by clustering observations according to their latent embeddings. The memory is utilized to more efficiently locate previously visited objects, particularly in long-horizon tasks. The paper also proposes an exploration-exploitation strategy allowing agents to alternate between exploration and task-solving based on recalled events. Experiment results showed that the proposed method outperforms traditional method (short-term memory and traditional FIFO memory) in long-horizon tasks.

**Strengths:**

1. The paper is technically sound and clearly describes the algorithm. The proposed approach (Place Event Memory, PEM) efficiently manages memory by integrating place-based and event-based memory, effectively reducing redundancy while preventing confusion from similar observations at different locations. This mechanism enables agents to recall previous memories during long sequences of tasks, thereby enhancing task efficiency.
2. The results on long-horizon tasks are good.
3. The presentation of the paper is well-executed.

**Weaknesses:**

1. The work requires further improvement:
    - The authors refer to their proposed approach as “Place Event Memory” (PEM), where event memory is implemented by clustering according to the latent embeddings encoded by MineCLIP. However, many frames during the tasks do not constitute meaningful events. For instance, when searching for an object, it may take several steps to navigate in the environment. These observations may not be considered as events (and these observations are also not trained in MineCLIP, I’m uncertain about how they will be clustered). Therefore, it may not be appropriate to call it “event memory”, and there's still some unnessary memory storage.
    - For exploration, the authors use a visitation map and always select positions with the lowest visit frequency. This rule-based method is not efficient and does not leverage the knowledge saved in PEM. Why did the authors not consider using the same mechanisms for both exploration and exploitation?
    - The experimental results require further explanation. In Section 4.4, place-based memory achieves better performance on long-instruction tasks, while event-based memory performs better on long-navigation tasks. This result seems counterintuitive. Can the authors further explain these experimental outcomes?
   - The key idea of this work is to record the positions of visited objects, framing the task as a lifelong navigation problem. There has been significant research on navigation, particularly regarding navigating in unseen environments and exploring to locate target objects while recording previously seen ones (e.g., building a semantic map). Why did the authors not include a comparison with these works?
2. symbol expression: On page 4, the notation $O_t=\{o_t, l_t, t\}$ is used. In Markov Decision Processes (MDPs), uppercase $O$ typically represents the observation space, while lowercase $o_t$ represents an element within that space. This notation should be clarified.
3. spelling errors: There are some spelling errors in the appendix. For example, on page 22, “Evene Cluster Details” should be corrected to “Event Cluster Details.”

**Questions:**

See Weaknesses

---

> ### Author Response · Authors · 2024-11-25
>
> We thank the reviewer for highlighting the strengths of our work, including the technical soundness, clear algorithm description, and strong performance in long-horizon tasks.
>
> ## Many frames in PEM do not constitute meaningful events and PEM may have unnecessary memory storage
>
> Thank you for highlighting this important aspect of our work. We appreciate your observations about the potential limitations of PEM. In PEM, it first segments the agent’s trajectory into place-based clusters, where each cluster contains observations from nearby locations and similar directions. This ensures that the each place cluster is spatially organized and includes observations relevant to specific locations. Then, event clusters are formed from the place cluster by using MineCLIP representation. However, as the reviewer mentioned, many frames in event clusters during the tasks do not constitute meaningful events. We agree that the term “event memory” might imply high-level semantic events, which is not always the case here. Our focus was on defining an event memory suitable for low-level controllers, which can store both semantic high-level events (e.g., burning zombies) and visually novel frames (e.g., cow in the forest) that might still be useful for future tasks. While this approach may include some redundant scenes, we opted for a structure that maximizes the chances of retaining frames potentially useful for subsequent tasks. Moreover, we recognize that optimizing memory to further reduce redundancy without losing important frames is a worthwhile direction for future work. To this end, we implemented a simple version where only center embeddings of event clusters are retained, which did not degrade performance significantly. Details are provided in **Appendix P**.
>
> We recognize that referring to the clustered experience frames as “event memory” may not align perfectly with the traditional meaning of “events.” The current version, based on place-observation memory, aligns more closely with the concept of What-Where-When memory as suggested in the title. In future versions, we plan to replace “event memory” with a more appropriate term, “3W-Memory.” However, to avoid causing confusion with the existing content, we have decided not to apply this change yet and will update it in the camera-ready version.
>
> ## Rule-based Exploration does not leverage knowledge in PEM
>
> Thank you for your constructive feedback. We acknowledge that the current hierarchical exploration in Mr.Steve selects positions with lowest visit frequency from visitation map, which does not utilize knowledge stored in PEM. We notice that task-conditioning is crucial for effective exploration. For instance, the agent should prioritize exploring a forest rather than a desert when searching for a tree. To address this, we have developed an advanced hierarchical exploration strategy that incorporates the same mechanisms used in exploitation. Further details are provided in Common Response **CR2**.
>
> ## Further explanations on Long-Horizon tasks in Section 4.4
>
> Thank you for your constructive feedback. We agree that the experimental results in Section 4.4 require further explanations, and we appreciate the opportunity to elaborate.
>
> In the **Long-Instruction task**, the agent is continuously assigned random “Obtain $X$” tasks, where $X$ could be water, beef, wool, log, dirt, or seeds. Resources such as beef and wool are located in visually similar forest-like areas but at different places. Mr.Steve and Steve-PM effectively retain task-relevant information for these distinct locations, whereas Steve-EM clusters visually similar events from different places into the same event cluster, potentially losing task-relevant frames. Consequently, Mr.Steve and Steve-PM solve over 80 tasks, while Steve-EM solves only around 50 tasks.
>
> In the **Long-Navigation task**, the task consists of exploration phase of $16$K steps and a task phase. In the exploration phase, the agent observes six events in different places: 1) burning zombies, 2) river, 3) sugarcane blow up, 4) spider spawn, 5) tree, and 6) house, spending $2$K steps at each place. In the task phase, the image goal is continuously given randomly selected from the frames in the early steps of the event. Note that event 1, 3, 4 are dynamic events which only occur in the early steps when the agent arrives at the event-occuring places. When the agent is in task phase, Mr.Steve and Steve-EM can maintain all events in the memory through event clustering (We note that each events are visually distinct enough for event clustering), while Steve-PM, with its FIFO-based place clusters, loses early steps of dynamic events. As a result, Mr.Steve and Steve-EM solved around 70 tasks, while Steve-PM solved less than 20 tasks.

---

> > ### Author Response · Authors · 2024-11-25
> >
> > ## Why navigation methods from other domains are not compared?
> >
> > Thank you for your valuable feedback regarding the navigation policy. We answered how our navigation methods relate to other domains such as robotics in Common Response **CR3.**
> >
> > ## Minor Updates: Change Notation for Observations and Typo Correction
> >
> > We have addressed the issue with the symbol expression by updating the notation for observations and pixel observations to  $X_t$  and $i_t$, respectively, to improve clarity. Additionally, we have corrected the spelling errors in the manuscript, including changing “Evene Cluster Details” to “Event Cluster Details.”

---

> > > ### Comment · Reviewer_UxTw · 2024-11-26
> > > **Thank the authors**
> > >
> > > I appreciate the authors’ response and the revisions made to the manuscript, which have addressed most of my questions, and I have updated my score to 6. However, I still believe the method lacks a comparison between PEM and SLAM-based approaches, particularly in terms of high-level subgoal selection for memory exploitation.

---

> ### Author Response · Authors · 2024-11-27
>
> We sincerely thank you for your thoughtful feedback and for updating the score. We have initiated a comparison study between PEM and SLAM-based approaches as suggested. While we are working diligently to complete these additional experiments, the time constraints of the extended discussion period may make it challenging to obtain results before the deadline. Nevertheless, we will make every effort to conduct this analysis and will update you with our findings as soon as they become available. We greatly appreciate your constructive comments, which have helped strengthen our work significantly.

---

> > ### Author Response · Authors · 2024-12-04
> >
> > We sincerely thank the reviewer for their valuable feedback, and for patiently waiting for our additional experiments. As suggested, we have conducted a comparison study between the PEM and a SLAM-based approach, focusing on their effectiveness in high-level subgoal selection for memory exploitation. The results of these experiments are reported in Common Response **CR4**.

---

### Official Review · Reviewer_UwZe · 2024-11-02

**Soundness:** 4
**Presentation:** 4
**Contribution:** 2
**Rating:** 6
**Confidence:** 4

**Summary:**

Authors present Mr.Steve, an extension of the Steve-1 instruction-following agent, improving exploration and memory abilities in Minecraft settings. The core contribution of authors is the design of a Place Event Memory system, which allows creating a hierarchical memory: a set of map checkpoints is constructed from the agent trajectory through clustering, and in each checkpoint multiple events are memorized. The memory is built and queried based on CLIP embeddings from a previous work, allowing to compute similarities between a language or visual instruction and image-based memories. Additionally, authors finetune an existing transformer-based trajectory embedding network to learn a goal-conditioned policy and implement a count-based exploration system. Authors conduct multiple experiments to showcase the performance of the overall system, which significantly outperforms Steve-1

**Strengths:**

This work tackles an important subject of research: how to design efficient language-conditioned agents in complex environments.

The paper is very well written. Experiments are in-depth and clearly described. Results are impressive.

I enjoyed reading it.

**Weaknesses:**

The current version of the paper could be more pedagogical in some parts.
For instance: How is are the text and video encoder aligned ? More explanation about this should be featured in the paper (i.e. more explanation about MineClip I guess).

Likewise, some details regarding the DP Means algorithm, used to figure out checkpoint locations, could be useful. E.g. how does it compare to the well known K-Means ? Does DP Means selects autonomously the number of clusters ? If no, how is the system detecting how many clusters/checkpoints to create for a given agent trajectory ?

l.243 "This structure improves the efficiency of the read operation by extracting top-k place clusters with their center embeddings first, then fetching relevant frames from these clusters"
--> This hierarchical decomposition assumes the “center embedding” is sufficient to figure out which FIFO memory to read. This looks like a strong assumption, e.g. the center embedding could correspond to the agent looking away from the object of interest. But maybe it is enough if clustering is well done ?

### Novelty/Significance

While very well executed, this work only tackles a Minecraft scenario, on a few individual contributions are proposed and efficiently combined: a hierarchical memory system, a count based exploration mechanism, and goal-conditioned navigation agent. Count based exploration and xy_goal-directed navigation are well known areas. I am not an expert regarding memory systems for decision-making agents, but similar systems might have been proposed in the past. I am also not expert enough to assess whether the considered baselines are sufficient.

I am looking forward to the discussion period to update my score, but from this first review I recommend acceptance, despite my aforementioned concerns. This work efficiently showcases how to combine and scale known components in a complex and relevant setting.


minor:

l.100 "(Hafner et al., 2023; Guss et al., 2019; Cai et al., 2023a; Mao et al., 2022; Lin et al., 2021; Zhou et al., 2024a)"
--> make sure for any multi-citation to order by year from old to new.

**Questions:**

see above

---

> ### Author Response · Authors · 2024-11-25
>
> We would like to extend our sincere thanks to Reviewer 29NL for their positive and encouraging feedback on our manuscript. Your constructive comments have been instrumental in refining our paper.
>
> ## More Explanations on MineCLIP
>
> As suggested, we added “, which is a CLIP model trained on web videos of Minecraft gameplay and associated captions.” in Line 232 in the updated manuscript. Thanks for the suggestion.
>
> ## Details on DP-Means Algorithm
>
> Thanks for the constructive feedback. We acknowledge that some details with DP-Means algorithm is insufficient. DP-Means algorithm is a Bayesian non-parametric extension of the K-means algorithm based on small variance asymptotic approximation of the Dirichlet Process Mixture Model. It doesn't require prior knowledge on the number of clusters $K$. To run this algorithm, we first set the initial number of clusters $K'$ and cluster the data with K++ Means initialization ($K'$ can be $1$), then DP-Means algorithm automatically re-adjust the number of clusters based on the data points and cluster penalty parameter $\delta$. Thus, DP-Means algorithm behaves similarly to K-means with the exception that a new cluster is formed whenever a data point is farther than $\delta$ away from every existing cluster centroid. We added the details of DP-Means **in Appendix E.** Event Cluster Details (**Line 1145**).
>
> ## Using center embedding for the first top-k cluster memory read
>
> Thank you for pointing out this aspect. We proposed one method for structuring the memory system, but it is possible to enhance this by setting multiple center embeddings per cluster (e.g., using an additional K-Means) to handle queries. While this would increase computational cost linearly with the number of center embeddings, the efficiency of the query operation can still be maintained as long as the number of center embeddings remains significantly smaller than the total data points within a cluster. In our experiments, we found that using a single center embedding per cluster was sufficient.
>
> Regarding your specific concern, we appreciate your insightful example. In our approach, we assume that the agent’s position and orientation are known, which helps ensure that each cluster in the place memory contains observations from nearby locations and similar directions (More details in **Appendix E**, **Line 1158**). However, as the reviewer pointed out, a single center embedding might not fully represent the agent’s focus within a specific space. This is because a single space can host various events, and in place memory, the center embedding is selected as the observation closest to the geometric center of a cluster. To address this limitation, our proposed Place Event Memory (PEM) incorporates event-based clustering within a single spatial area. By doing so, the center embedding for each cluster corresponds to a distinct and semantically meaningful event, improving the relevance of the retrieved information.
>
> ## Unclear Novelty and Significance of the paper
>
> Thank you for your thoughtful review and for recognizing the strong execution of our method. Below, we address the concerns about novelty and significance.
>
> **Count-based Exploration and Goal-conditioned Navigation** We acknowledge that count-based exploration is a well-established method, and we adopted it based on prior works in SLAM. For goal-conditioned navigation, we want to emphasize differences of VPT-Nav from previous works. In Steve-1, and GROOT [1], VPT is finetuned for goal-conditioned behavior cloning (supervised learning). In DECKARD [2], and PTGM [3], VPT is finetuned with adapter [4] for single task with reward (RL). We found that naively combining goal-conditioning from Steve-1, and RL finetuning from DECKARD showed suboptimal navigation behavior. Thus, we came up with different conditioning and recently proposed LoRA adaptor for RL finetuning. This resulted in VPT-Nav’s optimal navigating behaviors outperforming goal-conditioned navigation policy from Plan4MC (RL from scratch). We believe this approach provides insights into improving goal-conditioned navigation for foundation policy models. Further details are provided in **Appendix L**.
>
> **Memory System** In systems like Steve-1 that uses low-level controllers based on Transformer-XL architectures, recalling observations from more than a few thousand steps earlier is highly inefficient. To address this limitation, we proposed a Place Event Memory (PEM) that efficiently stores novel events from visited places, enabling sparse and effective sequential task-solving.

---

> > ### Author Response · Authors · 2024-11-25
> >
> > While memory systems are also employed in LLM-based Minecraft agents [5, 6], these systems typically store plans of successfully completed tasks with agent’s experience frames. Unlike our PEM, they lack mechanisms to efficiently store or utilize experience frames that are irrelevant to the current task but potentially useful for future tasks. PEM introduces a fundamentally different structure by enabling the storage and retrieval of such frames, ensuring that task-agnostic experiences are preserved for later use. Further details, along with examples, are provided in **Appendix A**.
> >
> > **Significance** The main focus of our work is addressing the critical issues in Steve-1, which serves as the standard low-level controller for nearly all Minecraft LLM agents with keyboard-mouse action space. Resolving these issues is highly significant, as it directly impacts the broader field of embodied AI. Additionally, to make the paper stronger, we advanced Mr. Steve by augmenting LLM for long-horizon high-level planning tasks (Common Response **CR1**) and task-conditioned hierarchical exploration (Common Response **CR2**).
> >
> > ## Minor Updates: Order of Multi-Citation
> >
> > We fixed the multi-citation order by year in the manuscript. Thanks for pointing out.
> >
> > **References**
> >
> > [1] Shaofei Cai, *et al*. “GROOT: Learning to follow instructions by watching gameplay videos.” ICLR. 2024.
> >
> > [2] Kolby Nottingham, *et al*. “Do Embodied Agents Dream of Pixelated Sheep?: Embodied Decision Making using Language Guided World Modeling.” ICLR Workshop on Reincarnating Reinforcement Learning. 2023.
> >
> > [3] Haoqi Yuan, *et al*. “Pre-Training Goal-based Models for Sample-Efficient Reinforcement Learning.” ICLR. 2024.
> >
> > [4] Neil Houlsby, *et al*. “Parameter-efficient transfer learning for nlp.” PMLR. 2019.
> >
> > [5] Zihao Wang, *et al*. “JARVIS-1: Open-world Multi-task Agents with Memory-Augmented Multimodal Language Models.” NeurIPS Workshop on Agent Learning in Open-Endedness. 2023.
> >
> > [6] Zaijing Li, *et al*. “Optimus-1: Hybrid multimodal memory empowered agents excel in long-horizon tasks.” Arxiv. 2024.

---

> > > ### Comment · Reviewer_UwZe · 2024-11-27
> > > **Answer**
> > >
> > > I thank authors for their response. Added details on MineCLIP and DP Means are helpful. The answer regarding the use of center embeddings is reasonable.
> > >
> > > Adding a full-loop appendix (using LLMs) is useful.
> > >
> > > At this point, I will keep my score (recommending acceptance).
> > > To increase further, given what I read from other reviews, I would like to see experiments of PEM vs SLAM-based approaches (see R-UxTw comments)

---

> > > > ### Author Response · Authors · 2024-12-04
> > > >
> > > > We thank the reviewer for continued support and for patiently waiting for our additional results. As suggested, we conducted a comparison study between the PEM and a SLAM-based approach. The results of these experiments are reported in Common Response **CR4**.

---

### Official Review · Reviewer_29NL · 2024-11-03

**Soundness:** 3
**Presentation:** 4
**Contribution:** 3
**Rating:** 8
**Confidence:** 4

**Summary:**

In the presented work, the authors propose an improvement over the widely used STEVE-1 low-level controller used in Minecraft challenges, addressing its limited episodic memory and inefficiency in handling long-horizon tasks. In particular, the authors propose Place Event Memory, a novel approach to storing "what-where-when" information. Utilizing this memory, the authors propose a new navigation policy, VPT-Nav, capable of balancing when exploration is needed and when direct task-solving can be done due to recalling prior information from the hierarchical PEM memory. Minecraft is a great testbed for dynamic environments and provides a challenging task for policies targeted at solving such settings where long-horizon task planning is needed.

**Strengths:**

- The introduction is well-written and intuitively understandable.
- The event-based clustering in the PEM memory seems novel an intuitive
- The results of Mr.Steve are strong and supportive of the claims made by the authors.

**Weaknesses:**

- In the PEM memory, clusters are updated based on some threshold c concerning the similarity of events/locations. However, it is unclear how this is handling events that happen dynamically. Even in the "burning zombies" example given in the PM section, this event would only happen when it's early morning in the game (thus night-zombies are burning), yet the PEM memory doesn't include the game time, meaning that such an event would be unreliable and there doesn't seem to be a way to capture this.
- The goal of Hierarchical Episodic Exploration seems to be to also prevent re-visiting places that have previously been seen. Given that the location of the agent is part of the place embeddings e_t, that seems feasible, however, if two environments are visually similar, that would mean that the agent would still explore as its location is a strong bias. However, this is a little in contrast to the goal of, for example, finding wood where searching yet another desert (just because the global position is different) would be suboptimal and should be avoided. So, in these settings, the global location would, on the one hand, need to be a strong separator to prevent searching the same environments, yet, at the same time, a weak separator because we wouldn't want to search the same biome for too long. How is this balance handled? I think a discussion on which factors exactly contribute to the creation of a new cluster would be beneficial for the paper.
- While the results in section 4.3 are supportive of the author's claims, it would be great if the same settings as in section 4.2 could be tested to demonstrate the impact of varying global memory limitations. E.g. which of the methods can solve the milk-sand-milk task with the least amount of global memory? Such a comparison would make the contribution much stronger.

**Questions:**

- In the example where meat needs to be found, what happens if navigating back to the location where cows were previously seen at does not have cows anymore? Is there a way to update/forget information in the PEM?
- In Figure 5, are the results, particularly comparing Mr.Steve, Steve-EM, Steve-PM, and Steve-FM, statistically significant (particularly for Wool-Dirt-Wool and Milk-Sand-Milk)?

---

> ### Author Response · Authors · 2024-11-25
>
> We express our gratitude to the reviewer for the recognition of novelty of the method and the strong results of our experiments.
>
> ## Can Event Clusters in PEM Capture Dynamic Events?
>
> Yes. In PEM, each of newly created clusters from DP-Means are either merged to existing event clusters or allocated as new event cluster for each place cluster. Whenever additional $100$ frames are stored in the place cluster, DP-Means [1] is applied and clusters are created. As the reviewer mentioned, each created cluster is merged to one of event cluster if the similarity of MineCLIP [2] representation of their center embeddings are higher than $c$, or allocated as a new event cluster otherwise. We used $c=73.5$ (in Appendix E.4) in all experiments for consistency.
>
> In burning zombies example, let’s assume zombies spawn at night in some place (confined in fence as in Figure 6(b)), and zombies burn and disappear in the morning. During night, PEM generates single event cluster since scenes during night are similar and resulting clusters from DP-Means are merged. However, when zombies burn in the morning, PEM generates new event cluster for burning zombies’ scenes, and another event cluster for scenes that zombies disappear. We found that MineCLIP representation works reasonably well to cluster semantically  different events with proper $c$. We also note that PEM includes the game time in the memory, as the experience frame $x_t= \\{ e_t,l_t,t \\}$ is stored in the memory as stated in section 3.1. Regarding this issue, we gave a more concrete answer in Common Response **CR0**.
>
> ## Balance between Strong and Weak Separator for Efficient Exploration
>
> We thank the reviewer for their insightful comments regarding the need for advancements on hierarchical episodic exploration. Following you feedback, we implemented the advanced version of high-level goal selector for more efficient exploration. See Common Response **CR2** for more details.
>
> ## Additional Experiment for ABA-Sparse Tasks with Memory Constraints
>
> We thank the reviewer for supportive feedback on ABA-sparse task in section 4.2. As suggested, we evaluated Mr.Steve, Steve-EM, Steve-PM, Steve-FM in 6 ABA-sparse tasks when there is a limitation in memory capacity. We tested different memory capacities (0.1K, 0.5K, 1K, 2K) for each model. In Figure 17 in Appendix O, we found that Mr.Steve maintained its performance even with low memory capacities. In case of Steve-FM, we observe that its performance decreases as the memory capacity gets small since it loses the experience frames for solving first task A. Interestingly, we found that Steve-PM, and Steve-EM showed degraded performance on Beef-Log-Beef and Beef-Leaves-Beef tasks when memory capacity is 0.1K. This indicates the robustness of PEM to memory capacities in ABA-Sparse tasks. For further details, we updated the paper with the results in **Appendix O**.
>
> ## In the example where meat needs to be found, what happens if navigating back to the location where cows were previously seen does not have cows anymore? Is there a way to update or forget information in the PEM?
>
> We answered this in Common Response **CR0**.
>
> ## In Figure 5, are the results, particularly comparing Mr.Steve, Steve-EM, Steve-PM, and Steve-FM, statistically significant (particularly for Wool-Dirt-Wool and Milk-Sand-Milk)?
>
> Thank you for the constructive feedback. We acknowledge the reviewer’s concerns regarding the statistical significance of the results in Figure 5. The purpose of the *ABA-Sparse* tasks in Figure 5 was to demonstrate the significant performance improvements of Steve-1 when augmented with memory. Upon reflection, we recognize that including ablations of Mr. Steve with different memory variants may not be good choice. Instead, we highlighted the benefits of PEM over various memory variants in **Section 4.3** and **Appendix O**, and added this content in Figure 5 caption.
>
> Additionally, we understand that the notation for Mr. Steve’s memory variants may have caused some confusion. Since these memory variants are all introduced in our paper, we will revise their names in the camera-ready version as follows: Steve-PM → Mr. Steve-PM, Steve-EM → Mr. Steve-EM, and Steve-FM → Mr. Steve-FM.
>
> **References**
>
> [1] Or Dinari, *et al*. “Revisiting DP-means: Fast scalable algorithms via parallelism and delayed cluster creation.” UAI. 2022.
>
> [2] Linxi Fan, *et al*. “Minedojo: Building open-ended embodied agents with internet-scale knowledge.” NeurIPS. 2022.

---

> ### Comment · Reviewer_29NL · 2024-11-26
>
> Thank you for taking the time to comment on my initial review. Most answers are compelling to me; however, the intuition behind the event clusters for dynamic events is still a little unclear to me. In your example detailing the burning zombies, I understand that PEM would generate new clusters as them burning and eventually disappearing would be a significant event. However, as far as I understand right now, this would require the agent to actually see the events to properly model them. In your task in which you need to retrieve items (e.g., Milk-Sand-Milk), if such a task includes a dynamic event (assume you want to find burning zombies), would MrSteve be able to anticipate certain locations? Assume two days ago, you saw that zombies burned in the morning at a particular location, but that location is far away now. Later on, you found, during last night, a location with zombies, but it was night, so they weren't burning (yet). Now, your task is to find burning zombies. Where would MrSteve go? The far-away zombies in the hopes of making it there in time (as you said, time is important for your clustering), or would it go to the significantly closer zombies that it hasn't seen burning yet but can transfer that knowledge from the faraway ones that the close ones will burn soon?

---

> ### Author Response · Authors · 2024-11-27
>
> Thank you for clarifying your question further. We now have a much clearer understanding of what you meant by 'dynamic event'. We think this is not a memory problem but rather an inference or prediction problem, where the agent needs to learn a world model that reflects the knowledge that night zombies turn into burning zombies in the morning. The current version of our PEM simply stores what it has observed, but does not fill the gaps between observations through reasoning. Thus, Mr.Steve would navigate to the far-away zombies which the burning is observed and not reaching to nearby zombies in your example.
>
> Nevertheless, we believe this question is very thought-provoking and interesting to consider for future work. One way to handle such a situation would be to have the LLM-based high-level planner read events from the PEM to reason about the fact that nearby night zombies will turn into burning zombies. We think this represents an interesting spatiotemporal reasoning challenge where GPT-like chain-of-thought reasoning could be helpful in the spatiotemporal space.
>
> Additionally, we could introduce temporal imagination capabilities similar to those used in planning models like Dreamer. Specifically, we could pick an event (e.g., night zombie) from PEM and use a learned world model to rollout its future state. If the model has learned effectively, it would predict the transformation into a burning zombie, which can let Mr.Steve navigate to nearby zombies in the example. In both approaches, however, this task seems more appropriate for the higher-level planner or reasoner rather than the low-level controller like Mr.Steve.

---

> ### Comment · Reviewer_29NL · 2024-11-27
>
> Thank you for the additional answer and thoughts on how your work could be used to make even better agents that can reason over the memory you are proposing in the current work. I fully recognize that this is out of the scope of this work; however, I think this would make for an interesting paragraph in future work. Either way, I will update my score to accept.

---

> > ### Author Response · Authors · 2024-12-04
> >
> > Thank you for the positive feedback and suggestions regarding potential future directions for our work. We are also grateful for your thoughtful review and for updating the score.

---

### Author Response · Authors · 2024-11-25

We appreciate the reviewers for their constructive feedback. We have revised our paper in response to the comments, with changes highlighted in color: **blue for 29NL**, **violet for UwZe**, **magenta for UxTw**, and **cyan for 86n7**.

## CR0. Adding more advanced functions to PEM
We sincerely appreciate the reviewers' thoughtful suggestions regarding additional functionalities for Place Event Memory (PEM). The reviewers have raised valuable points that will certainly help advance this line of research. In this paper, our primary objective was to introduce the novel concept of applying memory to low-level controllers and demonstrate its initial feasibility. As this represents the first exploration of this approach, we focused on developing and validating a foundational proof-of-concept implementation. We believe this initial step was crucial to establish the groundwork for future extensions and improvements of the PEM structure. Regarding the suggested features, such as temporal event segmentation (**Reviewer 29NL**), and memory update/forget mechanism (**Reviewer 29NL and 86n7**), we anticipate these functionalities could be implemented through the following approaches.

**Temporal Event Clustering**  PEM stores time information in the memory, which enables straightforward implementation of time-based clustering. This means that even if events are visually similar, we can effectively distinguish between them based on their different occurrence times.

**Memory Forget Mechanism**  Currently, among the task-related event clusters, the agent selects and moves to the location of events that occurred closest to its current position. However, since each event cluster also contains time information, we can refine the memory querying policy to prioritize more recent events. For instance, when the agent returns to a location where it previously observed cows and finds no cows present, a new memory is created with a corresponding time index. While the previous memory of seeing cattle still exists, its time index indicates that it occurred in the past. If there is another area where cattle were observed more recently, the agent can utilize this temporal information to navigate to that location instead.

**Memory Update Mechanism**  Some entities in Minecraft can change their state over time. However, the current PEM only memorizes observations in the past and lacks the ability to infer the current state from them. If we use world models, which have the ability to model uncertainty and predict future states of entities, it could be possible to implement the memory update mechanism. World Models like Recurrent State-Space Models (RSSM) [1] could be integrated to update MineCLIP representations in PEM by enabling it to predict future entity states based on the frame embedding. This approach would allow the memory system to handle the dynamic nature of the Minecraft environment.

Lastly, we would like to emphasize once again that our paper’s primary focus is to introduce minimal memory modules that can overcome the limitations of our main baseline, Steve-1. As this paper has successfully demonstrated the potential of this approach, we plan to explore more sophisticated memory structures in our future research.

## CR1. Extending Mr.Steve to LLM-based Agents
We are grateful for the insights provided by reviewers UwZe, and 86n7 on the applicability and scalability of our methods in general-purpose embodied AI scenarios. While the core contribution of our study is to introduce novel memory system in low-level controllers for embodied AI, and demonstrate its benefits when combined with simple exploration method, we agree that exploring more complex and realistic settings would make the paper stronger. Thus, we conducted additional experiments on combining LLM with Mr.Steve for tasks that require high-level planning to solve. We updated the paper with the results in **Appendix M**.

Specifically, when the task instruction is given (e.g., “make a bed”), LLM generates the text-based subgoals which are then given to the goal-conditioned low-level controller for execution. As a backbone LLM high-level planner, we employed DEPS [2]. Here, we tested on 4 tasks from Minecraft as shown in the following table, and compared two low-level controllers, Mr.Steve and Steve-1.

| Task | DEPS with Steve-1 | DEPS with Mr.Steve |
| --- | --- | --- |
| oak_stairs | 67% | **80%** |
| sign | 53% | **60%** |
| fence | 40% | **50%** |
| bed | 27% | **50%** |

---

> ### Author Response · Authors · 2024-11-25
>
> From the above table, we found that LLM-based agent shows higher success rates when using  Mr.Steve as low-level controller in all tasks. This is because most of the tasks require doing the same sub-task multiple times (e.g., mining a log three times), and memory in Mr.Steve can efficiently memorize the previous task and re-do it again. More elaborations and results can be found in **Appendix M**. Also, we provide the video of LLM-based Agent with Mr.Steve that solves ‘make bed’ task in https://imgur.com/a/brKrQhL.
>
> ## CR2. Demonstrating Task-Conditioned Hierarchical Episodic Exploration
> We sincerely appreciate the valuable insights provided by 29NL, UxTw and 86n7 regarding episodic exploration methods. Building upon these insights, we implemented and evaluated a task-conditioned hierarchical episodic exploration method that leverages knowledge stored in PEM.
>
> In task-free hierarchical episodic exploration (method proposed in the original manuscript), it selects the next goal randomly from least-visited locations. This does not utilize knowledge stored in PEM. For instance, the agent should prioritize exploring a forest rather than a desert when searching for a tree. To tackle this, task-conditioned hierarchical episodic exploration selects the next goal with highest task-relevance from least-visited locations. Specifically, task-relevance map is constructed to represent the task-relevant scores on agent’s trajectory. We used MineCLIP to calculate the task-relevant scores for each event cluster by computing the alignment score between center embedding of each cluster and text prompt.
>
> To see whether task-conditioned exploration benefits by using knowledge in PEM, we evaluated Mr.Steve with both exploration methods (task-conditioned and task-free) on three *ABA-Sparse* tasks. In the below table, success rate and agent’s exploration time are given for two exploration methods. While the performance improvement from task-conditioned exploration was not substantial, it significantly reduced exploration time by approximately 300~400 steps compared to the task-free method, indicating a more efficient exploration. For further clarity, we have updated the implementation details and full experimental results in **Appendix N.**
>
> |  | Task-Conditioned Exploration | Task-Free Exploration |
> | --- | --- | --- |
> | Beef-Log-Beef | 93% (1202.64$\pm$67.49) | 92% (1512.00$\pm$62.37) |
> | Wool-Sand-Wool | 98% (924.98$\pm$56.85) | 93% (1350.00$\pm$78.80) |
> | Milk-Leaves-Milk | 59% (1224.24$\pm$68.79) | 62% (1579.35$\pm$64.61) |
>
> ## CR3. What are the advantages of the proposed exploration method compared to those in the robotics domain?
>
> We are grateful for the insights provided by reviewers UxTw, and 86n7 regarding the relation between our proposed exploration method and those in robotics domain. We agree that the hierarchical exploration method we propose could be seen as simplified version of SLAM techniques commonly used in robotics. However, when comparing methods from robotics with ours, it is important to consider the distinction between high-level goal selector and low-level goal-conditioned navigation policy.
>
> At the high level, SLAM-based approaches typically use a top-down map [3] or construct a topological map [4, 5, 6] of the agent’s trajectory to propose the next exploration location or the optimal path between two locations. For instance, Robohop employs foundation models like SAM and DINO to create topological maps, and this method could complement our high-level Count-based exploration strategy.
>
> At the low level, however, the goal-conditioned navigation policies in these approaches often rely on RL policies trained from scratch [3] or built-in point navigation controllers for real robots [3, 5]. RoboHop, in particular, uses pixel-level heuristic navigation, which may not be suitable for complex and interactive environments like Minecraft. In scenarios where agents face challenges such as being pushed by enemies, crossing rivers or mountains, or navigating around sheep and cows, massive prior knowledge of the environment is essential for successful navigation. To address this, we proposed VPT-Nav, a goal-conditioned navigation policy built on VPT, a foundation policy model trained with human demonstration data. By applying adjustments such as LoRA adaptors and optimizing the placement of goal conditioning, we achieved significant improvements over previous VPT fine-tuning methods. VPT-Nav also outperformed RL navigation policies trained from scratch and heuristic policies used in Plan4MC, as detailed in **Appendix L.** We believe our approach to combining foundation policy models with goal-conditioned fine-tuning can be effectively leveraged with high-level goal selector in robotics.

---

> > ### Author Response · Authors · 2024-11-25
> >
> > **References**
> >
> > [1] Danijar Hafner, *et al*. “Dream to Control: Learning Behaviors by Latent Imagination.” ICLR. 2020.
> >
> > [2] Zihao Wang, *et al*. “Describe, explain, plan and select: Interactive planning with large language models enables open-world multi-task agents.” NeurIPS. 2023.
> >
> > [3] Matthew Chang, *et al*. “GOAT: GOto AnyThing.” ArXiv. 2023.
> >
> > [4] Kim Nuri, *et al*. “Topological Semantic Graph Memory for Image-Goal Navigation.” CoRL. 2022.
> >
> > [5] Hao-Tien Lewis Chiang, *et al*. “Mobility VLA: Multimodal Instruction Navigation with Long-Context VLMs and Topological Graphs.” CoRL. 2024.
> >
> > [6] Sourav Garg *et al*. “RoboHop: Segment-based Topological Map Representation for Open-World Visual Navigation.” ICRA. 2024.

---

> > > ### Author Response · Authors · 2024-12-04
> > > **Comparison between PEM and SLAM-based Method**
> > >
> > > ### CR4. Comparison between PEM and SLAM-based Method
> > >
> > > We sincerely appreciate the reviewer’s thoughtful suggestion regarding the comparison between PEM and SLAM-based approaches. To address this, we conducted experiments comparing PEM with Neural Topological Map (NTM) from Neural Topological SLAM (NTS) [1], a popular method among topological SLAM approaches. We found that metric-based SLAM methods [2,3] require a depth map, which our method does not utilize, so we focused on a topological method.
> > >
> > > NTM works as follows. NTM starts with a graph with a single node (agent’s node), and the node has pixel observation at initial time step. When the next pixel observation is given, NTM computes similarities between the next pixel observation and pixel observations from nodes in the topological graph. We used cosine similarity between MineCLIP representations for computing similarities. If the maximum of similarities do not exceed the threshold, we create a new node in the graph, and this node and agent’s node are connected by an edge that stores relative position. Then we update the agent’s node to a new node. If the maximum of similarities exceeds the threshold, we consider two cases. If the maximum similarity node coincides with the agent’s node, we update nothing. Otherwise, we connect the agent’s node with the maximum similarity node, and update the agent’s node to the maximum similarity node. Additionally, pixel observation in the agent’s node is updated with the current pixel observation. The biggest difference between NTM and PEM is the memory removal strategy. PEM retains diverse places and distinct events within each place by removing experience frames from the largest event cluster, whereas NTM retains only the most recent nodes in the graph, removing the oldest node.
> > >
> > > For evaluation, we tested NTM and PEM on 5 ABA-Sparse Tasks. Memory capacity was limited to 0.1K, where PEM demonstrated advantages over its variants (e.g., EM, PM). As shown in the table below, PEM outperforms NTM in most tasks. This is because NTM removes the oldest node from the graph, causing it to lose task-relevant knowledge of the initial task A while solving task B, which leads to lower success when revisiting task A at the end. We think that, if our understanding of NTM is correct, adapting NTM as PEM by modifying the memory write and removal operations may yield comparable results to our method.
> > >
> > > | ABA-Sparse Tasks with Memory-Constraints | PEM | NTM  |
> > > | --- | --- | --- |
> > > | Beef-Log-Beef | 0.94$\pm$0.03 | 0.70$\pm$0.05 |
> > > | Beef-Leaves-Beef | 0.97$\pm$0.02 | 0.73$\pm$0.04 |
> > > | Wool-Sand-Wool | 0.98$\pm$0.01 | 0.97$\pm$0.01 |
> > > | Milk-Sand-Milk | 0.85$\pm$0.04 | 0.75$\pm$0.03 |
> > > | Milk-Leaves-Milk | 0.79$\pm$0.02 | 0.46$\pm$0.07 |
> > >
> > > **References**
> > >
> > > [1] Devendra Singh Chaplot *et al*. “Neural Topological SLAM for Visual Navigation.” CVPR. 2020.
> > >
> > > [2] Devendra Singh Chaplot *et al*. “Learning To Explore Using Active Neural SLAM.” ICLR. 2020.
> > >
> > > [3] Matthew Chang, *et al*. “GOAT: GOto AnyThing.” ArXiv. 2023.

---

### Meta-Review · Area_Chair_anKB · 2024-12-22

**Metareview:**

The paper proposes to use language-based episodic memory to guide RL in instruction-following tasks. The paper is well motivated and presented, the evaluations are sufficient, and the results show clear impact. Reviewers raised some concerns of clarity that have mostly been addressed by the authors, however concerns regarding evaluation remain, namely the focus on navigational tasks may limit significance and calls for comparison with more classic methods for this setting.

**Additional Comments On Reviewer Discussion:**

Reviewers and authors engaged in a thorough discussion, resulting in good mapping of the paper's strengths and weaknesses.

---

### Decision · Program_Chairs · 2025-01-22

Accept (Poster)